# A genome assembly for *Orinus kokonorica* provides insights into the origin, adaptive evolution and further diversification of two closely related grass genera

Kunjing Qu[1,5], Ai Liu[1,5], Mou Yin[1], Wenjie Mu[1], Shuang Wu[1], Hongyin Hu[1], Jinyuan Chen[2,3], Xu Su[2,3], Quanwen Dou[4] & Guangpeng Ren [1✉]

Ancient whole-genome duplication (WGD) or polyploidization is prevalent in plants and has played a crucial role in plant adaptation. However, the underlying genomic basis of ecological adaptation and subsequent diversification after WGD are still poorly understood in most plants. Here, we report a chromosome-scale genome assembly for the genus *Orinus* (*Orinus kokonorica* as representative) and preform comparative genomics with its closely related genus *Cleistogene*s (*Cleistogenes songorica* as representative), both belonging to a newly named subtribe Orininae of the grass subfamily Chloridoideae. The two genera may share one paleo-allotetraploidy event before 10 million years ago, and the two subgenomes of *O. kokonorica* display neither fractionation bias nor global homoeolog expression dominance. We find substantial genome rearrangements and extensive structural variations (SVs) between the two species. With comparative transcriptomics, we demonstrate that functional innovations of orthologous genes may have played an important role in promoting adaptive evolution and diversification of the two genera after polyploidization. In addition, copy number variations and extensive SVs between orthologs of flower and rhizome related genes may contribute to the morphological differences between the two genera. Our results provide new insights into the adaptive evolution and subsequent diversification of the two genera after polyploidization.

[1] State Key Laboratory of Herbage Improvement and Grassland Agro-Ecosystems, College of Ecology, Lanzhou University, Lanzhou, China. [2] Key Laboratory of Biodiversity Formation Mechanism and Comprehensive Utilization of the Qinghai-Tibet Plateau in Qinghai Province, Qinghai Normal University, Xining 810008, China. [3] Academy of Plateau Science and Sustainability, Qinghai Normal University, Xining 810016, China. [4] Key Laboratory of Adaptation and Evolution of Plateau Biota, Northwest Institute of Plateau Biology, Chinese Academy of Sciences, Xining, China. [5] These authors contributed equally: Kunjing Qu, Ai Liu. ✉email: rengp@lzu.edu.cn

Whole-genome duplication (WGD) or polyploidization has been suggested to be prevalent throughout the evolutionary history of plants, acting as an important evolutionary force promoting diversification, speciation, and adaptation to new environments[1–4]. Neofunctionalization or subfunctionalization of duplicated genes post-WGD, as well as species-specific gene retention and loss, lead to expansion and contraction of gene families, resulting in the generation of morphological and adaptive diversity[5–7]. However, the process of species differentiation after ancient WGD is often accompanied by the occurrence of genome rearrangement events, resulting in the generation of new genes or changes in the expression and regulation mode of existing genes, which may result in the emergence of new traits and promote speciation[8,9]. Based on the origin and level of ploidy, polyploidy can be divided into two types, autopolyploidy and allopolyploidy. The former involves duplication of the diploid genome in a single species; whereas the latter is formed by a genomic combination between two or more distant species[3,10]. Although the relative proportion of the two types are comparable, allopolyploids are generally expected to have higher evolutionary advantages due to the pairing of chromosomes from each parent, which often results in more chromosomal structural variations, novel gene interactions, and morphological innovations[3,11]. In general, polyploid plants exhibit greater vigor, stronger resistance, wider environmental adaptability, and higher biomass and economic yield than their diploid relatives[8,11–13]. However, the underlying genomic basis of ecological adaptation and subsequent diversification remain poorly understood in most polyploid species.

Chloridoideae, containing approximately 124 genera and 1602 species, is the fourth largest subfamily of the grass family (Gramineae)[14]. Most species of this subfamily use the C4 photosynthetic pathway, and the first evolutionary transition from C3 to C4 photosynthesis was found in this subfamily about 32–25 million years ago (Ma)[15]. Species of this subfamily possess strong tolerance to arid environments[14], such as *Oropetium thomaeum* and *C. songorica*[16,17]. With a base chromosome number of $x = 10$, it is suggested that more than 90% of species within this subfamily are polyploids, including the staple grain crop teff in Ethiopia and many turfgrass and forage species[18]. Ancient WGD events may have contributed to the stress tolerance, emergence of valuable traits, and diversification of these grasses[17,18]. Thus, this subfamily may serve as a valuable system for investigating potential impacts of WGD, subsequent subgenome divergence, functional innovations, genome rearrangement, and diversification in polyploid plants. Recent advances in sequencing technologies have facilitated genome assemblies for polyploid species. To date, 16 reference genomes of this subfamily have been reported, 13 of which are polyploid. Five genomes were used for comparative examination of desiccation tolerance and sensitivity[19,20]. Whereas the rest mainly focused on the genomes themselves and investigated the divergence between subgenomes or the genomic basis of adaptation to extreme environments[16,17,21,22]. Among these studies, however, few have focused specifically on evolutionary divergence after polyploidization. Thus, a comprehensive understanding of how subsequent speciation and diversification occurs in these polyploid genomes remains unknown[9].

We focus on a newly named subtribe, Orininae, belonging to the tribe Cynodonteae, Chloridoideae[14,23]. This subtribe includes two genera, *Orinus* (3 species) and *Cleistogenes* (17 species), which are often difficult to separate morphologically. For example, Hao (1938)[24] has treated one *Orinus* species, *O. kokonorica* (K.S. Hao) Tzvelev, as a *Cleistogenes*. However, there are two distinct morphological differences between the two genera. *Cleistogenes* has hidden cleistogamous spikelets concealed within the upper sheaths, which is not found in *Orinus*; and *Orinus* has elongated rhizomes covered with leathery and glossy scales, while *Cleistogenes* has a cespitose habit or very short rhizomes[25]. *C. songorica* is an allotetraploid species ($2n = 4x = 40$) and possesses a genomic basis for its dimorphic flower differentiation and drought adaptation[17]. Our karyotyping also revealed a chromosome number of 40 for *O. kokonorica* (Fig. 1a), indicating that this species is a tetraploid plant. However, there is no reference genome for *Orinus*, which largely limits our understanding of its evolutionary history and genomic divergence between these two genera. Here, we selected *O. kokonorica* as a case study and performed comparative genomics between *O. kokonorica* and *C. songorica* to investigate genomic basis of adaptive evolution and diversification after polyploidization.

*Orinus kokonorica* is mainly distributed in the northeastern Qinghai-Tibet Plateau (QTP) at elevations between 2400 and 4200 m[26], whereas *C. songorica* widely occurs in semi-arid and desert areas in central Asia at relatively low altitude[25]. Both species are important forage resources with good palatability and high nutritious value for local livestock in arid regions. The two species display differences mainly in flower and rhizome morphology and in altitudinal distributions. Whereas dimorphic flower in *C. songorica* may have contributed to its survival and reproductive success under drought conditions[17], the elongated rhizomes in *O. kokonorica* may store and allocate nutrients for the plant growth and protect dormant buds underground for overwintering in the QTP. In addition, with elongated rhizomes, the species has developed a strong underground root system (Fig. 1a), which may not only contribute to its high drought tolerance, but also has high ecological significance for sand fixation and water conservation in alpine arid regions[25]. In this study, we first performed genome assembly and RNA-seq-based transcriptomics under different stress treatments for *O. kokonorica*. Then, using publicly available genomic and transcriptomic data for *C. songorica*, we conducted comparative genomic and transcriptomic analyses to elucidate the evolutionary history of the two species and the genomic basis of their subsequent divergence.

## Results

**Genome sequencing, size estimation, and assembly.** The genome size of *O. kokonorica* was estimated to be ~520 Mb by $K$-mer analysis based on 35.29 Gb of cleaned Illumina data (Supplementary Fig. 1a, Supplementary Table 1). A combination of Illumina, Nanopore, and Hi-C technologies were adopted for sequencing to accurately assemble its genome. Based on 61.87 Gb of Nanopore long reads corresponding to 110× coverage of the estimated ~520 Mb genome (Supplementary Table 1), we polished the raw assembled genome using NextPolish and performed deredundancy with purge_haplotigs, resulting in a final genome assembly with length of 556 Mb and contig N50 of 9.08 Mb (Supplementary Table 2). The Benchmarking Universal Single-Copy Orthologs (BUSCO) evaluation score was 97.6%, indicating a very complete and high-quality genome assembly (Table 1). Based on ~137 Gb of Hi-C data, we further connected 127 contigs onto 20 pseudochromosomes. In total, 99.80% (554.86 Mb) of the assembly was anchored and oriented on 20 pseudochromosomes (Supplementary Fig. 2, Supplementary Table 3). 91.37% of Illumina short reads could be properly paired mapped to the final genome assembly (Supplementary Table 4). These assessments indicate that the genome of *O. kokonorica* was assembled with high quality, completeness, and accuracy.

**Genome annotation and gene prediction.** A total of ~327 Mb (58.81%) of the *O. kokonorica* genome assembly was identified as repetitive elements. The vast majority of repeats were classified as

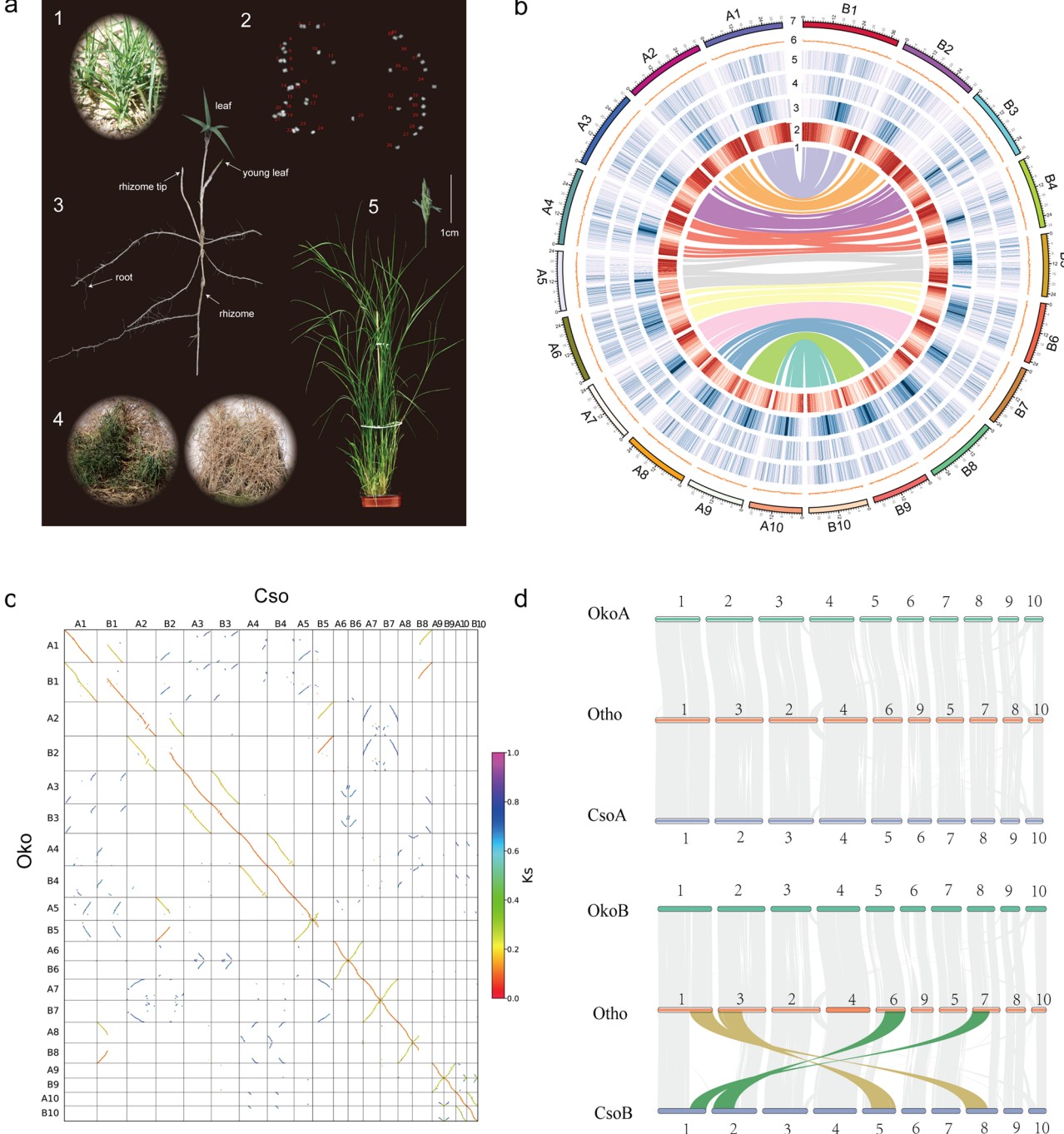

**Fig. 1 Characterization of the allotetraploid *O. kokonorica* genome. a** Morphology of *O. kokonorica*. 1: Plants in native habitat. 2: Chromosome counts (2*n* = 40) in an *O. kokonorica* cell. 3: Plant with leaves, roots and rhizomes. 4: Underground root system. 5: Plant grown in greenhouse. **b** Collinearity within the *O. kokonorica* genome. Tracks from inside to the outside correspond to (1) gene synteny between chromosomes; (2) gene density; (3) LTR/*Gypsy* retrotransposons; (4) LTR/*Copia* retrotransposons; (5) ncRNA density; (6) GC content; (7) Pseudochromosomes of *O. kokonorica*. **c** Dotplots showing the *K*s values of homologous genes between the Oko (*O. kokonorica*) and Cso (*C. songorica*) genomes. **d** Gene synteny between the Oko, Cso, and Otho (*O. thomaeum*) genomes.

LTR retrotransposons, accounting for 36.02% of the genome, including approximately 28.71% *Gypsy* and 7.30% *Copia* retro-transposons (Supplementary Table 5), a higher percentage than that in *C. songorica* (26.54%). Analysis of dynamic evolution of LTRs indicated that LTRs of *O. kokonorica* were younger than those in *C. songorica*, with the former experienced a more recent expansion with a peak of 0.8 Ma (Supplementary Fig. 3). DNA transposons, long interspersed nuclear elements (LINEs) and

short interspersed nuclear elements (SINEs) accounted for 8.92%, 3.56%, and 0.22%, respectively, of the genome assembly (Supplementary Table 5).

In total, a high-confidence set of 48,598 protein-coding genes was predicted using a combination of de novo, homology-based, and transcriptome-based approaches, and 48,521 (99.84%) were anchored into 20 pseudochromosomes (Table 1; Supplementary Table 6). With similar features of other Gramineae species,

**Table 1 Assembly and annotation summary of *O. kokonorica*.**

| Assembly features | |
| --- | --- |
| Genome size (bp) | 555'935'511 |
| N50 of contig (Mb) | 9.08 |
| N50 of scaffold (Mb) | 28.44 |
| GC ratio (%) | 45.51 |
| BUSCO score of assembly (%) | 97.60% |
| *Genome annotation* | |
| Number of protein-coding genes | 48,598 |
| Average gene/CDS length (bp) | 4327.41/1346.62 |
| Average exon/intron length (bp) | 217.93/575.54 |
| Average exons per gene | 6.18 |
| Repeats in genome (%) | 58.81 |
| BUSCO score of annotation (%) | 90.60% |

protein-coding genes in *O. kokonorica* were 4327 bp long and covered 6.18 exons on average. The lengths of exons and introns were highly conserved in all five investigated plant genomes (Supplementary Fig. 4, Supplementary Table 7), which further illustrated the reliability of the annotation results. In addition, BUSCO analysis of the protein set showed that the annotated genome contained 90.60% BUSCOs (Supplementary Table 8), suggesting good annotation completeness of protein-coding genes. Approximately 90.59% of *O. kokonorica* genes could be annotated by non-redundant nucleotides and proteins in the SWISS-PROT Protein Sequence Database, Gene Ontology (GO), Kyoto Encyclopedia of Genes and Genomes (KEGG), Clusters of orthologous groups for eukaryotic complete genomes (KOG) and Non-Redundant Protein Sequence Database (NR) (Supplementary Table 9). We also identified 231 microRNA (miRNA), 1012 small nuclear (snRNA) genes, 903 transfer RNA (tRNA), 183 ribosomal RNA (rRNA) in the genome sequence (Supplementary Table 10). Characterization and features of the *O. kokonorica* genome are exhibited in Fig. 1b. The LTRs were mainly distributed across the pericentric regions, while genes were mainly enriched in the more distal chromosomal regions.

**Allotetraploid origin of *O. kokonorica*.** The karyotyping $(2n = 4x = 40$, Fig. 1a), the $K$-mer analysis in which two peaks were present with the smaller peak at the doubled multiplicity of the major peak (Supplementary Fig. 1), and the chromosome-scale pairwise syntenic relationships within *O. kokonorica* genome (Fig. 1b, Supplementary Fig. 5) indicated that *O. kokonorica* was a tetraploid. To determine whether the species is an allotetraploidy or autotetraploidy, we first used Smudgeplot[27] to visualize the haplotype structure and to estimate ploidy of the genome. The result indicated an obvious peak of AABB type of $K$-mers (Supplementary Fig. 1b), implying the allotetraploid origin of *O. kokonorica*. This is further supported by the genome synteny and synonymous substitutions per synonymous site ($K$s) values between *O. kokonorica* and two *C. songorica* subgenomes (Fig. 1c). $K$s is assumed to be neutral, if *O. kokonorica* is an autotetraploidy, we would expect that the $K$s values between the two duplicated chromosomes of *O. kokonorica* and one of their two syntenic *C. songorica* chromosomes are similar. However, our result showed that each *C. songorica* chromosome corresponded with a pair of *O. kokonorica* pseudochromosomes with apparently different $K$s values, supporting the allotetraploid origin of *O. kokonorica*.

Based on genome synteny and $K$s values between *O. kokonorica* and the two *C. songorica* subgenomes, we divided the *O. kokonorica* pseudochromosomes into A and B subgenomes. It should be noted that the two subgenomes of *C. songorica* are simply determined based on the phylogenetic

distance between *C. songorica* and *Oryza sativa*[17], which should be treated with caution. Nonetheless, this will not affect our comparative genomic analyses because aligning the chromosomes of the two species is prerequisite for such analyses. We found that high collinearity among the two genomes by visualizing the synteny blocks. For subgenomes A of the two species, each pair of chromosomes corresponded very well, while for subgenomes B, two pairs of large interchromosomal rearrangements were revealed between chromosomes B2 and B5, and between B1 and B8. When using *O. thomaeum*, a diploid close relative to the two allotetraploid species, as a reference, we found no interchromosomal rearrangement events between *O. kokonorica* and *O. thomaeum* (Fig. 1d), indicating that these large interchromosomal rearrangements occurred specifically in *C. songorica*.

**Comparative genomic and phylogenetic analysis.** A comparative genomic analysis was performed among 16 plant genomes, including 12 Gramineae species, 1 Bromeliaceae species, 1 Musaceae species, 1 Orchidaceae species, and 1 Brassicaceae species. For the three allotetraploid species (*O. kokonorica*, *C. songorica*, and *Eragrostis tef*), their subgenomes were used. In total, 34,019 *O. kokonorica* genes (70.00%) were clustered into 25,903 gene families, of which 8961 families were shared with the other four Gramineae species (*O. sativa*, *C. songorica*, *Zea mays*, *O. thomaeum*) and 1387 were unique to the *O. kokonorica* genome (Supplementary Fig. 6a, Supplementary Table 11). These unique gene families were predicted to have functions involved in 'regulation of salicylic acid metabolic process' (GO:0010337), 'amino acid transport' (GO:0006865), and 'male-female gamete recognition during double fertilization forming a zygote and endosperm' (GO:0080173) (Supplementary Fig. 6b).

We identified 495 expanded and 6400 contracted gene families in the A subgenome of *O. kokonorica* compared to the other plant species, and the corresponding numbers in the B subgenome was 549 and 6556, respectively. However, compared with *O. kokonorica*, the number of expanded gene families in *C. songorica* was larger (1234 in A and 1350 in B subgenomes), whereas the number of contracted families was much lower (1645 in A and 1491 in B subgenomes, Fig. 2a, Supplementary Table 12). GO analysis revealed that the expanded orthogroups of *O. kokonorica* were significantly enriched in the functional terms 'programmed cell death involved in cell development', 'jasmonic acid biosynthetic process', and 'regulation of double-strand break repair' (Supplementary Fig. 7, Supplementary Data 2). KEGG analysis of these expanded gene families revealed significant enrichment in the "DNA repair and recombination proteins pathway", "flavonoid biosynthesis pathway", and "environmental adaptation pathway" (Supplementary Fig. 8, Supplementary Table 13).

The same 16 plant species were used to infer the phylogeny of *O. kokonorica* based on 381 single-copy orthologous genes. Subgenomes of the three allotetraploid species were used for the phylogenetic reconstruction and estimation of divergence times. As expected, *O. kokonorica* displayed the closest relationship with *C. songorica*. The diploid progenitors of subgenomes A and B (now likely extinct) of *O. kokonorica* and *C. songorica* diverged from their common ancestor ~18.19 Ma (12.99–24.31 Ma, Fig. 2a). The ancestor of *O. kokonorica* and *C. songorica* diverged from *O. thomaeum* ~ 20.86 Ma (15.21–27.33 Ma), which together diverged from *E. tef* ~ 26.19 Ma (19.82–31.33 Ma, Fig. 2a). The phylogenetic relationships among these 16 species were the same as those recovered from previous studies[22].

The divergence times estimated by phylogenetic and $K$s analyses between subgenomes of *O. kokonorica* and *C. songorica*

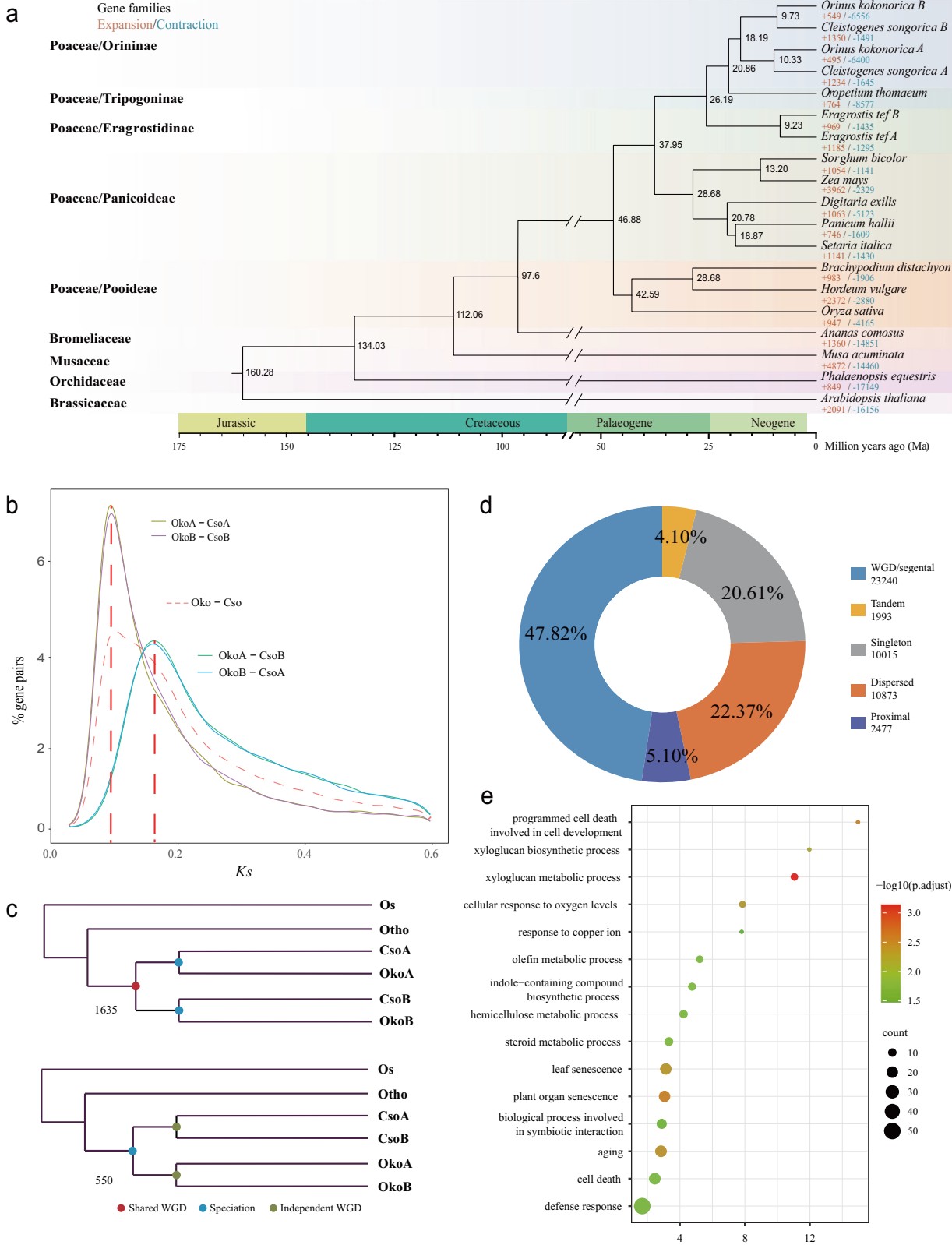

**Fig. 2 Evolution of the *O. kokonorica* genome. a** Phylogenetic relationships and divergence times among 16 species. The numbers at nodes represent the divergence times. Gene family expansion (+) and contraction (−) are indicated by red and blue, respectively. **b** Distribution of average synonymous substitution levels (*K*s) between syntenic blocks. **c** Two possible scenarios of species formation for *O. kokonorica* and *C. songorica*, with the number in the lower left corner of each tree representing the number of gene trees that support the topology (Os: *O. sativa*; Oko: *O. kokonorica*; Cso: *C. songorica*; Otho: *O. thomaeum*). **d** Classification of gene duplicate origins in the Oko genome. **e** Gene Ontology (GO) enrichment analysis of the tandem duplicated genes in Oko. Circle color represents -log(FDR) (false discovery rate) in the hypergeometric test corrected using BH (Benjamini and Hochberg) method. Circle size represents the gene count of the GO terms.

suggested that the two species (or the two genera) may have shared one paleo-allotetraploid event. The A and B progenitors were estimated to diverge from each other ~18.19 Ma, corresponding to the overlapped $K$s peaks between A-B subgenomes of the two species (OkoA–CsoB, OkoB–CsoA, Fig. 2b, Supplementary Data 1). The $K$s analysis also showed that $K$s peaks of OkoA–CsoA, OkoB–CsoB, and Oko–Cso overlapped, which indicated that the *Orinus–Cleistogenes* divergence, A subgenomes divergence, and B subgenomes divergence occurred at the same time, i.e., ~10 Ma (Fig. 2a). We further performed phylogenetic analysis to test this shared allotetraploidy event. Using the *O. sativa* genome as a reference, colinear genes of *O. thomaeum*, *O. kokonorica*, and *C. songorica* were extracted to build the gene trees. We found that most collinear gene trees (74.8%, Fig. 2c) well supported the shared allotetraploidy event between *O. kokonorica* and *C. songorica*. In other words, the two genera diverged from the same allotetraploidy ancestor, and the allotetraploidy event occurred between 18.19–10 Ma.

**Tandem duplication contributed to adaptation to harsh plateau environment**. Although *Orinus* and *Cleistogenes* shared the same paleo-allotetraploidy event, *O. kokonorica* has experienced severe gene loss (large number of contracted gene families) compared with *C. songorica* during its adaptation to the high plateau. We found only 47.82% (23,240) of gene duplications resulted from WGD/segmental duplication. There was a considerable number of duplicated genes generated from dispersed duplication (22.37%, 10,873), proximal duplication (5.10%, 2477) and tandem duplication (4.10%, 1993) (Fig. 2d). Tandemly arrayed genes are thought to be volatile after gene duplication, and therefore the retained tandemly genes may indicate functional importance[28]. Therefore, we investigated GO enrichment of the tandemly duplicated genes and their expressions under three stress treatments (cold, heat, and drought; see details in the Methods). We found that they were enriched in some GO terms related to adaptation to the plateau environment, such as 'cellular response to oxygen levels', and 'leaf senescence' (species at higher altitudes have longer leaf life than those at lower altitudes)[29] (Fig. 2e). Furthermore, 755 (37.9%) out of 1993 tandemly duplicated genes were differentially expressed in at least one tissue or treatment, and 236 (26.2%) out of the total 902 tandem gene clusters had at least two differentially expressed copies, suggesting the importance of tandem duplicated genes in adaptation to harsh environmental conditions.

**Genomic fractionation bias and subgenome dominance in *O. kokonorica***. Duplicated genes following polyploidization can be retained through subfunctionalization/neofunctionalization or reverted to a single copy through genome fractionation. We investigated genome fractionation in *O. kokonorica* and found that the two subgenomes of *O. kokonorica* retained a similar number of genes across the chromosomes, showing no biased fractionation (Supplementary Fig. 13).

Unequal expression of homoeologous genes in allopolyploids can be an important feature and consequence of polyploidization. We investigated homoeolog expression bias in five different tissues (mature leaf, young leaf, root, rhizome, and rhizome tip) and under three stress treatments (cold, heat, and drought; see details in the Methods). Of 11,861 identified homoeologous gene pairs between subgenomes A and B, 11,409 had homoeologous expression bias (HEB) in at least one tissue or treatment, and 1160 had biased expression in all sampled tissues and treatments. Pairwise comparisons between syntenic gene pairs showed a slight bias in expression toward the B subgenome across all five tissues and three treatments (Supplementary Fig. 10). Roughly

26.6% (3155) of the 11,861 pairwise comparisons across the five tissues and three stress treatments showed biased expression toward homoeologs in the B subgenome. Each tissue and treatment have ~6000 homoeologous gene pairs with differential expression, including 52.16% biased toward the B subgenome (Supplementary Fig. 10).

**Genome-wide variation and differentiation in homoeolog expression patterns between *O. kokonorica* and *C. songorica***. Genome-wide variation, including SNPs, indels, duplications, and structural variations (SVs), was identified based on whole-genome alignment with *O. kokonorica* and *C. songorica* genome assemblies. A total of 9,522,801 SNPs, 44,618 indels, 2768 duplications, and 925 inversions were identified using NUCDIFF (Fig. 3a). Based on Assemblytics, we identified a total of 41,559 SVs, in which presence/absence variants (PAVs) accounted for 23.78%. Most SVs were detected in non-coding regions, while 26.57% of the SVs were present in exon regions (Supplementary Fig. 9), which could have affected gene function and led to divergence between the two genera. A total of 14,604 genes were categorized as SV-high-impact genes (i.e., the SV is assumed to have a high or disruptive impact on the protein by causing protein truncation, causing loss of function, or triggering nonsense-mediated decay), which were mainly enriched in 'flower development', 'post-embryonic development', and 'regulation of gene expression, epigenetic', and involved in 'mismatch repair', 'cytochrome P450', and 'homologous recombination' pathways (Supplementary Fig. 9).

To investigate the differentiation in homoeolog expression patterns between *O. kokonorica* and *C. songorica*, we first performed transcriptomic analysis under the same stress treatments (i.e., light drought, cold, heat) in both shoots and roots of *O. kokonorica* as conducted for *C. songorica*. Hierarchical cluster analysis of transcriptomic data showed clustering of the four repeats for the shoot or root at each treatment, except for one root and shoot sample in the cold treatment group and one shoot sample in control group (Supplementary Fig. 11), which were discarded for downstream analysis. Differentially expressed genes (DEGs) were identified under different treatments by comparison with the control group. A total of 1246 and 2316 DEGs in shoots and roots, respectively, were identified under cold stress with 2-fold changes compared with the control sample. 338 DEGs in shoots and 420 DEGs in roots were identified under heat stress. However, under light drought stress, much lower DEGs (11 in shoots; 631 in roots) were detected compared to the other two treatments (Supplementary Fig. 12). The number of DEGs in roots were about 57 times greater than in shoots, indicating that drought stress in early stages does not considerably affect the shoots of *O. kokonorica*.

Next, we downloaded transcriptomic data under the same stress treatments for *C. songorica* and identified DEGs using the same method as described for *O. kokonorica*. It should be noted that although we have tried to keep the same environmental conditions as those reported in ref. [17], other factors (e.g., artificial factor and equipment performance) may generate effects on gene expressions. To reduce such effects, we only focused on the orthologous genes that were both DEGs in the two species (either in shoots, roots, or both) under each treatment and only examined the differential expression pattern of two orthologous genes in each pair. A total of 533, 99, 350 orthologous DEG pairs were identified under cold, drought, and heat treatments, respectively (Fig. 3b, Supplementary Data 3–5). We found that 53% (52) of orthologous genes had conserved functions (i.e., with the same differential expression pattern in the same tissue under the same treatment) in the two species under light drought

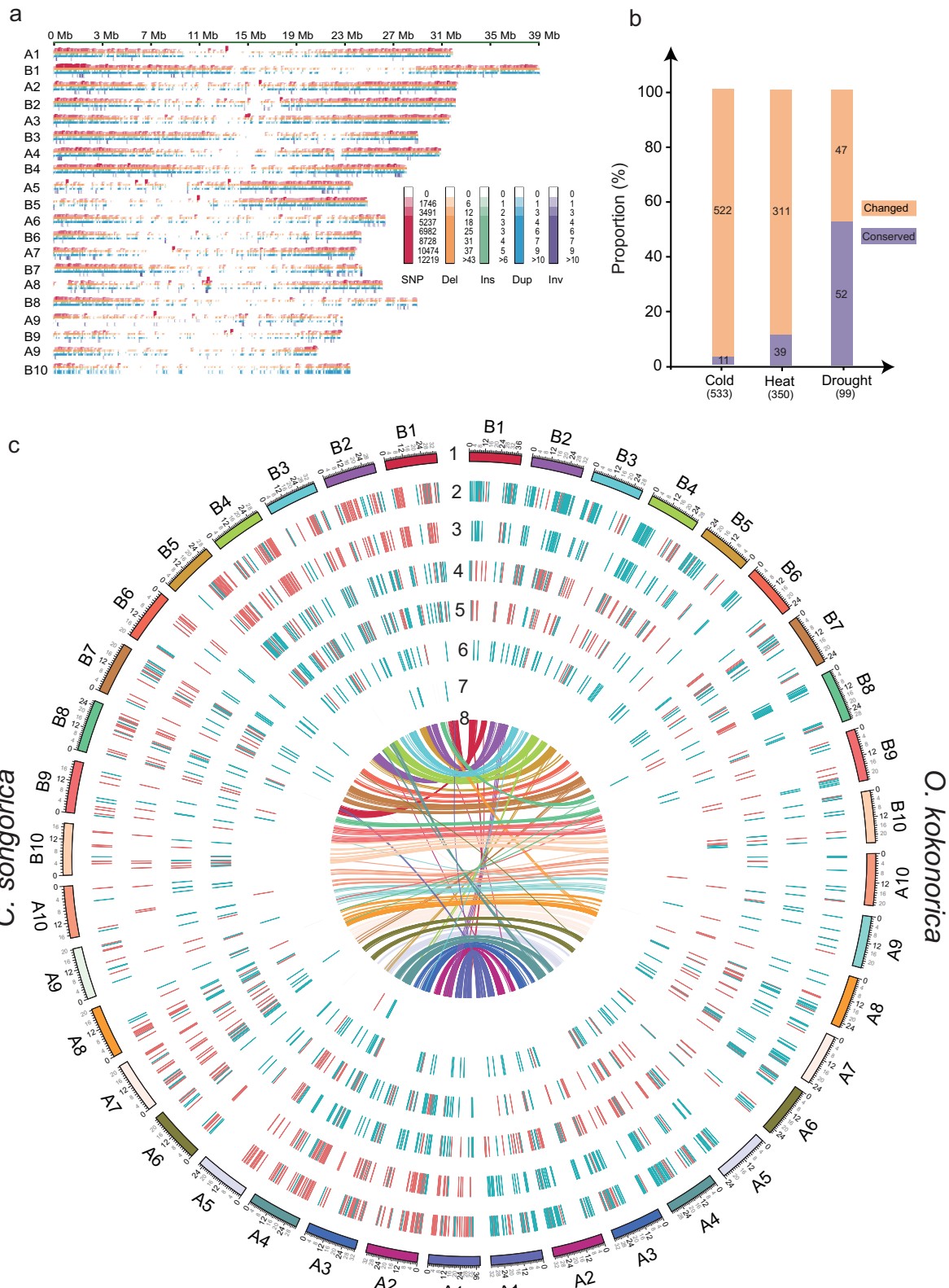

**Fig. 3 Genetic differentiation between *O. kokonorica* and *C. songorica*. a** Distribution of genetic variations on chromosomes. From top to bottom are SNP, deletion, insertion, duplication, and inversion. **b** Summary of expression patterns of the orthologous DEG (differentially expressed genes) pairs in *O. kokonorica* and *C. songorica* under cold, drought, and heat treatments. **c** Expression patterns of the orthologous DEG pairs in *O. kokonorica* and *C. songorica*. Tracks from outside to inside correspond to 1) chromosomes; 2) DEGs in root under cold treatment; 3) DEGs in shoot under cold treatment; 4) DEGs in root under heat treatment; 5) DEGs in shoot under heat treatment; 6) DEGs in root under drought treatment; 7) DEGs in shoot under drought treatment; 8) DEGs synteny between chromosomes. For (b–g), blue color indicates downregulation, red color indicates upregulation.

treatment, consistent with the fact that both are drought resistance species. However, only ~2% and ~11% of orthologous genes under cold and heat treatments, respectively, had the same expression patterns in the two species (Fig. 3b, c). Most orthologous genes were differentially expressed either in different tissues or in different ways (i.e., up- vs. down-regulation) or both, which may be related to the distinct altitudinal distributions of the two species.

**Genetic basis of differentiation in floral development and rhizome growth between *Orinus* and *Cleistogenes*.** Floral development is effectively explained by the ABCDE model[30], and a recent study has suggested that differential expression of the ABCDE genes may affect the development of the chasmogamous (CHs) and cleistogamous flowers (CLs) in *Amphicarpaea edgeworthii*[31]. Another study has revealed that overexpression of *CsAP2_9* gene confers an abnormal palea in a spikelet and shows degenerated lodicules in flowers in *C. songorica*[17]. Therefore, to gain some insights into why dimorphic flowers fail to develop in *Orinus*, we focused on the ABCDE genes that had differential expression in CLs compared with that in CHs in *C. songorica*[17] and compared them between *C. songorica* and *O. kokonorica*. We first examined and compared the ABCDE model genes (AMGs) in *O. sativa*, *O. kokonorica*, and *C. songorica*. Using 21 AMGs in *O. sativa* as references, we found that, after the shared allotetraploidy event, many copies were lost in *O. kokonorica* (31), while *C. songorica* gained more copies (46) (Fig. 4a). Second, based on transcriptomic data of *C. songorica* flowers, we found that 43.5% (20) of AMGs showed differential expression (twofold change) in CLs compared with that in CHs (Supplementary Table 14). Three orthologs of these AMGs (*MADS13*, *MASD16*, and *MASD7*) were lost in *O. kokonorica*, and multiple SVs, such as deletions or insertions upstream, intron, or even exon regions of genes, were detected in the remaining 17 AMGs between the two species (Fig. 4b, c). For example, two deletions were detected in the exon regions of *Oko006142* compared with its ortholog (*CsA201468*) in *C. songorica*. Another gene *Oko019761* gained two MYB-binding domains compared with its ortholog in *O. kokonorica*. These SVs detected in the orthologous AMG pairs may affect their expression and contribute to the differentiation of floral development in the two genera.

Rhizomes are modified stems that grow horizontally underground in various perennial species, a growth habit that is advantageous for vigorous asexual proliferation. *O. kokonorica* has elongated rhizomes, while *C. songorica* has a cespitose habit or very short rhizomes. The suppression of leaf blade development is necessary for the underground growth of rhizomes, and *BLADE-ON-PETIOLE* (*BOP*) homologs are important in the regulation of rhizome growth[32]. Based on *Arabidopsis. thaliana* and *O. sativa BOP* genes, we identified 7, 6, and 3 *BOP* genes in *O. kokonorica*, *C. songorica*, and *O. thomaeum*, respectively, and found that each *BOP* in *O. thomaeum* had synteny to two in the former two species, except one extra tandem copy in *O. kokonorica* (*Oko044410* and *Oko044411*, Fig. 5a, b). All 7 *BOP* genes were highly expressed in rhizomes and rhizome tips of *O. kokonorica*, especially the two copies of *BOP1* (*Oko010623* and *Oko007168*, Fig. 5c), which may promote the stiffness and sharpness of the rhizome tips in this species to further form a strong underground root system. In addition, we also detected extensive SVs in the *BOP* orthologous pairs between the two species (Supplementary Table 15).

## Discussion

In this study, we report the first chromosome-scale genome assembly of *O. kokonorica*, and the first for the genus *Orinus*. The

karyotyping result together with our comparative genomic analyses confirmed that *O. kokonorica* is an allotetraploid species, with all chromosomes confidently assigned to two subgenomes. Unlike most polyploid genomes in the grass family, the allotetraploid *O. kokonorica* has a relatively compact genome (~556 Mb), similar to other tetraploid species from the subfamily Chloridoideae, such as *C. songorica* (540 Mb)[17], *E. tef* (576 Mb)[21], and *Leptochloa chinensis* (416 Mb)[22]. The small genome size in these tetraploids is probably inherited from their diploid ancestors, such as that of *O. thomaeum*, a diploid species closely related to these four tetraploids, whose genome is only 245 Mb in size[16]. However, we found higher proportions of repetitive elements and a fewer number of genes in *O. kokonorica* than those in the other three species (Supplementary Table 16). The former is mainly resulted from the rapid expansion of LTR-RTs in *O. kokonorica* in the very recent past of approximately 0.8 Ma compared with *C. songorica* (Supplementary Fig. 3). This time frame is congruent with the largest Naynayxungla glaciation in the QTP, which reached its maximum between 0.8 and 0.6 Ma[33]. The severe environmental conditions during this period may have induced the bursts of TEs[34,35], which may have contributed to the increase in the genome size of this species. Nonetheless, the enormous contraction in gene families in *O. kokonorica* (Fig. 2a) may have counteracted this contribution to genome size. Therefore, the *O. kokonorica* genome might have been maintained by a reciprocal offset between recent expansion of TEs and contraction in gene families.

Our collinear gene phylogenetic and *K*s analyses indicated that the two closely related genera, *Orinus* and *Cleistogenes*, may share one paleo-allotetraploidy event between 18.19–10 Ma. This polyploidization event is older than the ones detected in their relatives *E. tef* (1.1 Ma)[21] and *L. chinensis* (<10.9 Ma)[22]. Following polyploidization, duplicated genes can be retained through neo-functionalization/subfunctionalization or reverted to a single copy through genome fractionation. Biased genome fractionation may lead to a dominant subgenome, which has often been reported in the genomes of allopolyploids especially in allopalaeopolyploids[36], such as in *Arabidopsis*, bread wheat, maize, and *Brassica rapa*[37–40]. However, we found that the two subgenomes of *O. kokonorica* have retained similar numbers of genes across the chromosomes (Supplementary Fig. 13) and display no significant global gene expression dominance, which is consistent with that found in *C. songorica*[17]. Such non-biased homoelog expression between subgenomes in allopolyploids seems not as rare as previously thought, as this pattern is also observed in other allopolyploid species, including *E. tef*[21], pumpkin[41], and *E. crus-galli*[42].

Our analyses of comparative genomics and transcriptomics provide some new insights into the genetic basis of divergence between the two genera after paleo-allotetraploidization. First, we found two pairs of large interchomosomal rearrangements occurred specifically in B subgenome of *C. songorica*. Although no gene changes were found at the breakpoints and these translocation regions have high collinearity between the *O. kokonorica* and *C. songorica* genomes, these large genomic rearrangements may affect chromatin organization as revealed in two ecotypes of *Medicago truncatula*[43], which may result in different epigenetic modifications and gene transcriptional activity in the two species[44]. Nevertheless, whether these genomic rearrangements are present in all *Cleistogenes* species and whether these translocations affect chromatin organization needs further investigation. Second, SVs constituted a large proportion of the genomic variation that results in phenotypic variation in organisms[45]. Using two methods, we built an overview of the genomic landscape of SVs between *O. kokonorica* and *C. songorica*, many of which may have contributed to the phenotypic variation and

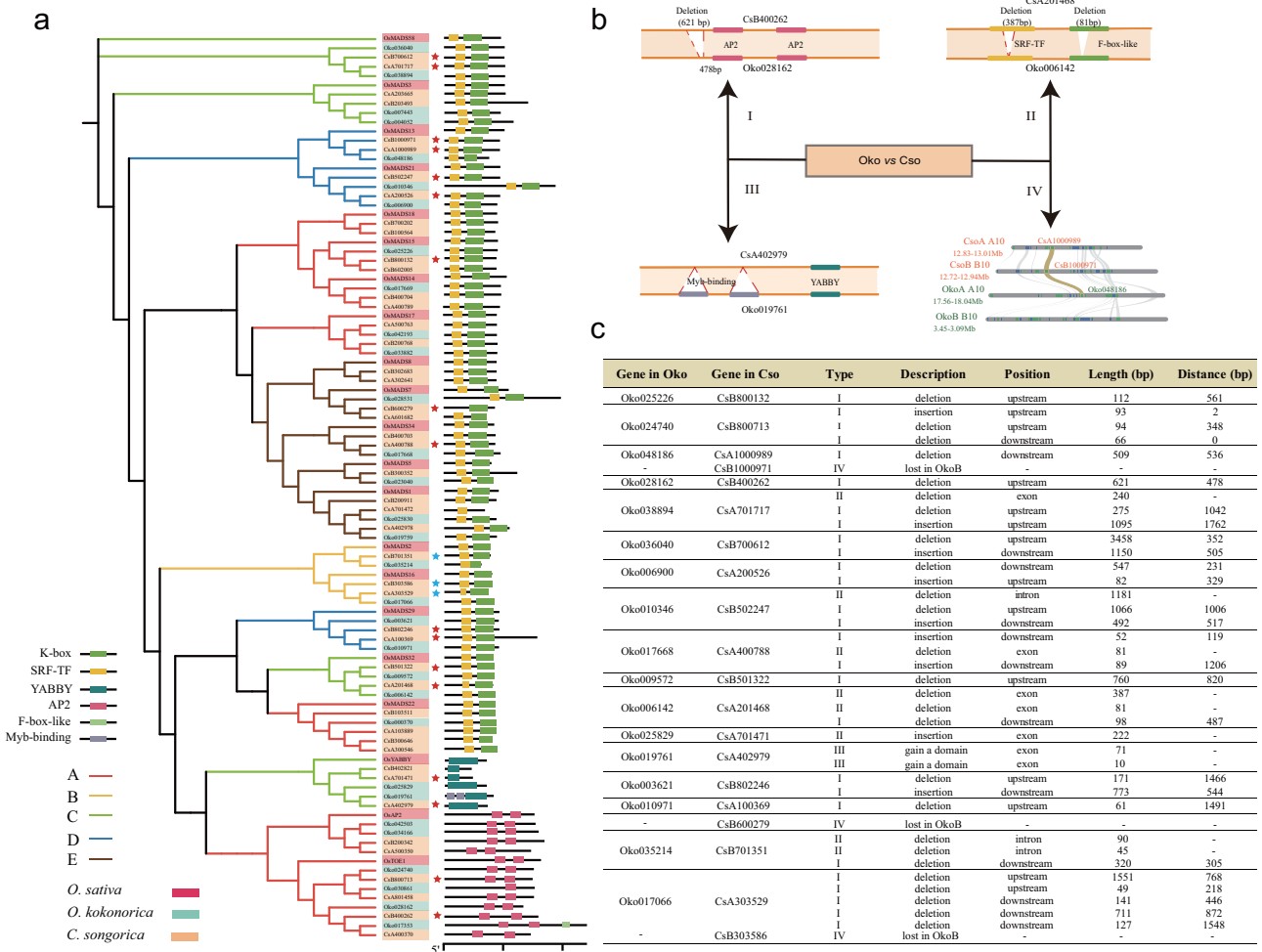

**Fig. 4 Structural variations (SVs) of the AMGs between *O. kokonorica* and *C. songorica*. a** The maximum likelihood phylogenetic tree of the AMGs identified in *O. kokonorica*, *C. songorica,* and *O. sativa*. Red stars represent the 17 AMGs with higher expression (twofold change) in CLs compared with that in CHs in *C. songorica*. Blue stars represent the 3 AMGs with lower expression (twofold change) in CLs compared with that in CHs in *C. songorica*. **b** Examples of the four types of SVs detected in the 20 orthologous AMG pairs identified in (**a**). (I) SVs in upstream or downstream regions of genes; (II) SVs in gene regions (including both intron and exon); (III) Domain gain or lost; IV) gene lost. **c** The detailed SVs in the 20 orthologous AMG pairs between the two species. "Type" are the SV types in (**b**).

diversification of the two genera. For example, some highly-impacted-by-SV genes were enriched in "flower development." We also found extensive SVs in some flower development and rhizome growth-related genes and their vicinity (<2 kb), which may play important roles in determining the morphological differentiation of flowers and rhizomes between the two genera (see more detailed discussion below). Finally, while both genera have strong resistance to drought, *Orinus* occurs in high altitude regions and has adapted to cold environments; *Cleistogenes* is tolerant to heat in low altitude regions[25]. Consistent with this, we found most orthologous DEGs (98% for cold treatment and 89% for heat treatment) in the two species under cold and heat treatments exhibited functional divergence by differential expression in different tissues or in different directions. By comparison, more than half of the orthologous DEGs under drought treatment have conserved functions in the two species (Fig. 3b). In addition, tandem copy increases in gene family members may have contributed to the adaptation of *O. kokonorica* to the plateau environment, as also indicated in many other organisms[46,47]. Therefore, genomic rearrangements, SVs and functional innovations of orthologous genes may have together played an important role in promoting genomic

divergence and speciation between the two genera after polyploidization.

Despite being closely related, *Orinus* and *Cleistogenes* have two distinct morphological differences. *Cleistogenes* has a dimorphic flowering mechanism, while *Orinus* has elongated rhizomes, both of which ensure reproductive success under harsh environmental conditions[48–50]. We investigated the genetic basis of differentiation in floral development and rhizome growth between the two genera. Using AMGs in *O. sativa* as references, we found that many duplicated AMG copies have been lost in *O. kokonorica*, while *C. songorica* has gained more copies. The increase in copy number of this gene family followed by functional innovations might have played an important role in the dimorphic flower development in *Cleistogenes*[17,51]. We further focused on the genes with differential expression in CLs than in CHs in *C. songorica*. We first found one copy of *MADS13*, one copy of *MASD7* and one copy of *MASD16* were lost in *O. kokonorica* (Fig. 4, Supplementary Table 14). All the three genes are suggested to be involved in determining floral organ identities in *O. sativa*[52–54]. The differential expression in CLs compared with that in CHs of the three gene copies in *C. songorica* may contribute to the development of CLs. Furthermore, we detected extensive SVs

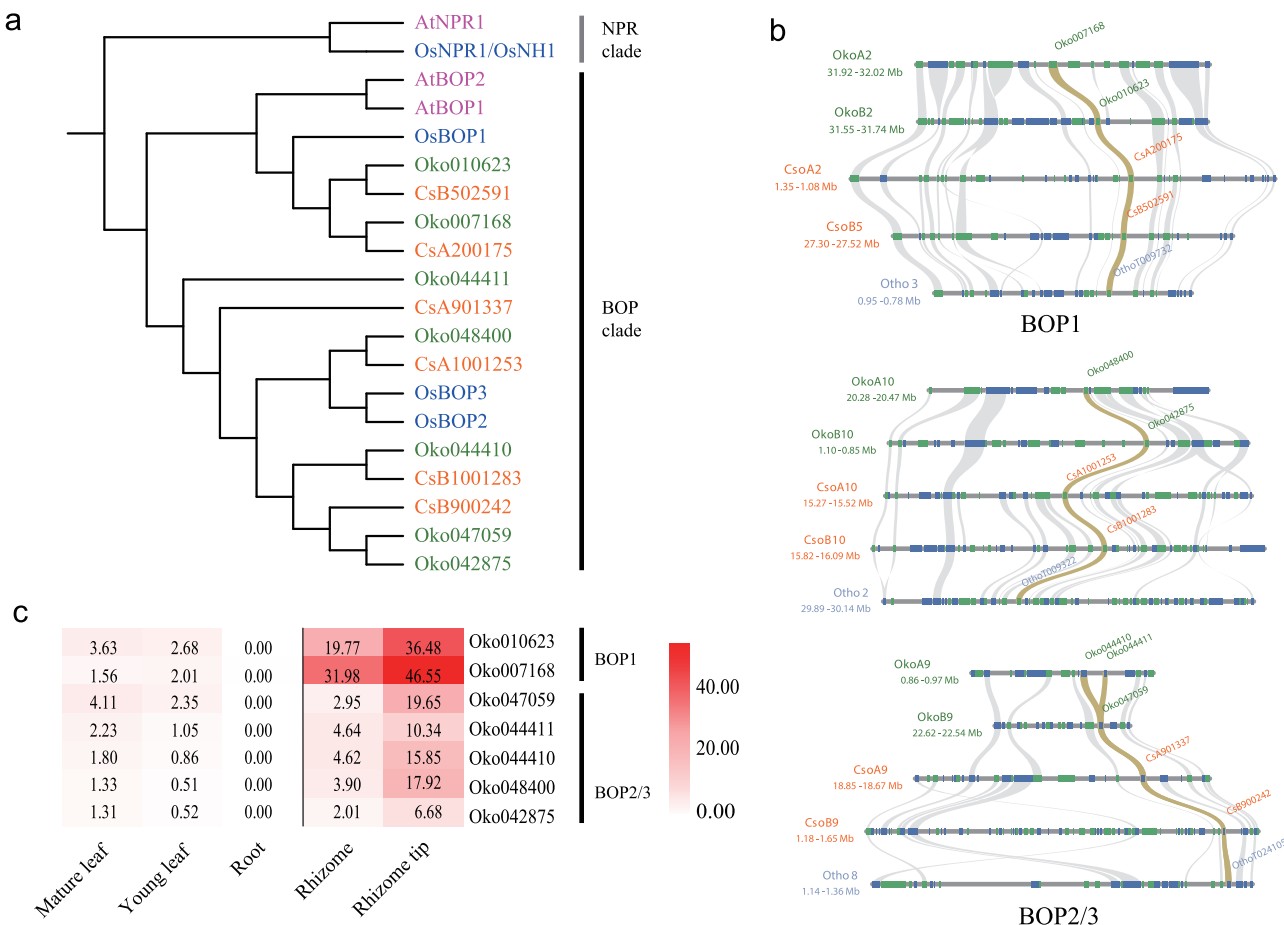

**Fig. 5 Characterization of BOP Genes in O. kokonorica. a** A phylogenetic tree based on BOP/NPR protein sequences identified in *O. kokonorica, C. songorica, O. sativa,* and *A. thaliana.* The *NPR* genes are shown as outgroup. **b** Collinear gene blocks between *O. kokonorica* and *C. songorica* indicate that the *BOP* family has one more tandem duplicated gene in *O. kokonorica*. **c** The expression of the *BOP* genes in five tissues of *O. kokonorica.* Expression level is indicated by transcripts per million (TPM).

between the orthologous pairs of these 20 DEGs, especially in the regulatory regions (Fig. 4c). SVs in regulatory regions or introns may result in dosage variations in gene expression, which has been shown to play a crucial role in the variation of plant traits, especially for floral organ identity[55,56]. Many studies have suggested that changes in expression of a single ABCDE-class gene can easily shift the boundaries between different types of floral organs[57,58]. Therefore, extensive SVs in the AMGs may have played an important role in the differentiation of flower development between the two species. Similarly, we detected more copies of *BOP* genes that are all highly or moderately expressed in rhizomes of *O. kokonorica* and extensive SVs between *BOP* orthologous genes in *O. kokonorica* and *C. songorica*. The copy number variation and SVs of *BOP* genes may have contributed to the differentiation of rhizome length in the two species. However, our present data are not appropriate to gain detailed mechanistic explanations of the two morphological differentiations. Further studies involving more comparative genomic analyses associated with experimental validation would greatly enhance our understanding of the molecular mechanisms of flower and rhizome development in the two genera.

## Methods

**Plant materials, karyotyping, and estimation of genome size.** The plant materials of *O. kokonorica* were collected from east side of the Qinghai Lake, Qinghai province, China. Bleach-sterilized seeds were germinated in a glasshouse. Root tips with a length of 1–2 cm were harvested from germinated seeds and pretreated in ice-cold water at 4 °C for 20–24 h. Root tips were then fixed in ethanol: glacial acetic acid (3:1, v/v) for 4 h at room temperature. Each root tip was squashed in a drop of 45% acetic acid and observed using a phase contrast microscope (Olympus Bx40). The slides containing metaphase chromosomes were frozen at −80 °C for more than 30 min and cover slips were removed with a razor. Chromosomes were counterstained with 4',6-diamidino-2-phenylindole. Images were captured with a cooled charge-coupled device camera using a fluorescence microscope (Olympus Bx63).

Fresh leaves of *O. kokonorica* were used to extract genomic DNA using a DNA Secure Plant Kit (Tiangen Biotech, Co., Ltd., Beijing, China). Paired-end libraries with insert sizes of 150 bp were constructed and sequenced using the Illumina HiSeq X Ten platform. We used *K*-mer frequency distribution analysis to estimate genome size[59]. To calculate and plot the *K*-mer frequency distribution, we used 35.29 Gb of Illumina short reads (Supplementary Table 1) to determine the total number of *K*-mers of length 27 using Jellyfish v. 0.9.0[60]. The selection of 27-mers to plot the frequency distribution was based on genome characteristics and the pattern of the Poisson distribution (Supplementary Fig. 1).

**Genome sequencing and assembly.** Total genomic DNA was further fractionated into 10–50 kb fragments with BluePippin to construct libraries following the Nanopore library construction protocol. The libraries were sequenced at the Nextomics

Biosciences Company (Wuhan, China) using the GridION X5 sequencer platform (Oxford Nanopore Technologies, UK). Quality-controlled reads were used for de novo assembly using Nextdenovo v. 2.3.051 (https://github.com/Nextomics/NextDenovo). We corrected sequencing errors with NextCorrect, assembled with NextGraph, and polished with NextPolish v. 1.2.424[61]. At this stage, the short and long reads were used three times for genome correction. Finally, purge_haplotigs[62] was used to remove allelic haplotigs, resulting in the final genome assembly. Benchmarking Universal Single-Copy Orthologs (BUSCO) v. 2.026[63], with 1614 genes from Embryophyta odb10, was used to evaluate the completeness and accuracy of the assembled genome.

**Chromosome-scale assembly with Hi-C data**. Hi-C libraries were constructed from ~2 g of fresh leaves following[64], which included chromatin extraction and digestion and DNA ligation, purification, and fragmentation. Then, these libraries were submitted for sequencing using the Illumina HiSeq X Ten platform (Illumina, CA, USA). The generated clean Hi-C data were mapped to the draft genome using BWA v. 0.7.1753[65]. The uniquely mapped Hi-C data were retained, clustered, ordered, and placed onto the 20 pseudochromosomes using LACHESIS[66]. A heat map of the interaction matrix of all pseudochromosomes was plotted with a resolution of 100 kb.

**Genome annotation**. Repetitive elements, including tandem repeats and transposable elements (TEs), were predicted in the *O. kokonorica* genome based on a combination of homology-based and de novo approaches. Tandem repeats were annotated using Tandem Repeats Finder v. 4.09[67]. TEs were identified using RepeatMasker v. 4.1.055 and RepeatProteinMask[68] with Repbase v. 22.11[69] as the query library. Next, RepeatModeler v. 1.0.1.0[68] was used to construct a de novo repeat library for the identification of TEs that were not found in the Repbase library. We used LTR_FINDER[70] to search the full-length long terminal repeat (LTR) retrotransposons in *O. kokonorica*, *C. songorica*. We extracted the 5′ and 3′ LTR sequences based on the LTR_FINDER annotation results, which were subsequently aligned using MUSCLE v. 3.8.31[71]. The genetic distance (K) between the 5′ and 3′ LTR sequences was calculated using DNADIST, a program within PHYLIP v. 3.696[72]. The insert time (T) of each LTR-RT was calculated using the formula: $T = K/2r$, where r is the nucleotide substitution rate estimated by BASEML (see below).

The repeat masked genome was used for predicting subsequent protein-coding genes with a combination of three complementary methods: de novo, homology-based, and transcriptome-based prediction. Augustus v. 3.3.257[73], GlimmerHMM v. 3.0.458[74] and Genscan[75] were used for de novo predictions. GeMoMa v.1.3.161[76] was used for homology-based predictions, with protein sequences from *A. thaliana*, *Eragrostis curvula*, *E. tef*, *O. thomaeum*, *O. sativa*, *Prunus persica*, *Sorghum bicolor*, *Triticum aestivum*, *Z. mays*. For transcriptome-based predictions, we first sequenced the RNA libraries generated from five tissues (i.e., root, rhizome, rhizome tip, young leaf, and mature leaf) and assembled the RNA-seq reads into transcripts using Trinity v. 2.1.162[77] with default parameters. We also used all the RNA-seq reads to assess genome assembly quality by mapping to the final assembled genome using PASA v. 2.1.063[78] Finally, all predictions of gene models yielded by the above approaches were integrated using EVidenceModeler (EVM) v. 1.1.1[79] to generate a consensus gene set.

The predicted protein-coding genes were functionally annotated by searching against databases. We used Interproscan v. 5.36[80], including Gene Ontology (GO) database annotations,

protein motifs and domains, functional classifications, protein family identification, transmembrane topology, and predicted signal peptides, to obtain a comprehensive annotation of the predicted protein-coding genes. We used a custom Perl script to get the annotation information. Then, KOBAS (http://kobas.cbi.pku.edu.cn/annotate/) was used to search the Kyoto Encyclopedia of Genes and Genomes (KEGG) database[81] for orthologs. Finally, we used BLASTP to search against the Swiss-Prot[82], NR[83], and KOG databases with an e-value cutoff of 1e−5. All of the best hits of these database searches were integrated to obtain the final functional annotation result.

Transfer RNAs (tRNAs) were identified by tRNAscan-SE[84] with eukaryotic parameters. miRNA and snRNA genes were predicted using INFERNAL[85] by searching against the Rfam database[86]. rRNA genes were predicted by aligning reads to *Arabidopsis* template rRNA sequences using BLASTN with an e-value of 1e−5.

**Polyploidization analyses and subgenome identification**. We combined three approaches for polyploidization analyses. First, Smudgeplot was used to visualize the haplotype structure and to estimate ploidy of the genome. Second, protein sequences within and between genomes were aligned using BLASTP with an e-value cutoff of 1e−5. Syntenic blocks were detected based on homologous gene pairs using MCScanX[28] and MCScan (https://github.com/tanghaibao/jcvi/wiki/MCscan-[Python-version]). Finally, WGD and speciation events were inferred by calculating *Ks* values using MCScanX downstream analysis program (Perl). *O. kokonorica* and *C. songorica* are both tetraploids belonging to the same subtribe of Chloridoideae, and *C. songorica* has been revealed to be an allotetraploid species with 20 chromosomes assigned to two subgenomes[17]. Therefore, we partitioned the *O. kokonorica* genome into subgenomes A and B based on dotplot with syntenic blocks and synonymous nucleotide substitutions (*Ks*) values against the *C. songorica* subgenomes using WGDI[87].

We further used the collinear gene phylogenomic analysis to check if WGD events are shared between *O. kokonorica* and *C. songorica*. Collinear genes were extracted by WGDI with "-at" parameters and the maximum likelihood trees were inferred using IQ-TREE[88] with automatic selection of the best-fit substitution model by (-m MFP) and 1000 ultra-fast bootstrap replicates (-bb 1000). The frequency of gene trees supporting shared WGD event was calculated using ASTRAL[89]. For example, in two species A and B (assuming each has one WGD event), the collinear genes extracted in the two species were named as A1, A2, B1, and B2. Thus, the topology supporting shared WGD event in A and B is ((A1, B1), (A2, B2)), while the topology of ((A1, A2), (B1, B2)) supporting independent WGD events in A and B. We used the ASTRAL with "-t2" parameters and the specific gene tree ((A1, B1), (A2, B2)) to calculate the number of gene trees that support shared WGD event.

**Gene families and phylogenetic analysis**. OrthoMCL was used to identify the orthologous groups among 12 Gramineae species (*O. kokonorica*, *C. songorica*, *O. thomaeum*, *E. tef*, *S. bicolor*, *Z. mays*, *Digitaria exilis*, *Panicum hallii*, *Setaria italica*, *Brachypodium distachyon*, *Hordeum vulgare*, *O. sativa*), two Commelinids species (*Ananas comosus*, *Musa acuminata*), one Monocot species (*Phalaenopsis equestris*) and one Rosid species (*A. thaliana*). For the three tetraploids (i.e., *O. kokonorica*, *C. songorica*, and *E. tef*), both subgenomes were used for orthologous group construction and phylogenetic analysis. The dynamic evolution of gene families in these 16 species was predicted using CAFÉ v. 3.1[90], and the significantly expanded or contracted gene families were

determined based on p-values ($p < 0.01$). We then completed GO enrichment and KEGG analyses on the expanded gene families in *O. kokonorica*.

Single-copy orthologous genes from the orthologous clustering results were extracted and aligned using MAFFT v. 7.158b[91]. Then, Gblocks v. 0.9171[92] was used to delete regions with poor alignment or large differences after multiple alignments. A maximum likelihood phylogenetic tree was reconstructed based on the single-copy orthologous gene data set using RAXML v. 8.1.17[93] with the PROTGAMMAJTTX model and 1000 bootstrap replicates. Nucleotide substitution rate and divergence time were calculated by four-fold degenerate sites (4DTv) of single-copy orthologous genes. The nucleotide substitution rate was estimated using BASEML v. 4.8a, a program within PAML v. 4.0[94]. The program uses a maximum likelihood method with tree file with topology in Newick format and DNA sequence alignment in Phylip format as input files to calculate an average nucleotide substitution rates over the entire tree. Species divergence time was inferred by MCMCtree in the PAML program, based on the calculated substitution rate and known approximate divergence times for *A. comosus* and *M. acuminata* (102–120 Ma), and *A. thaliana* and *P. equestris* (148–173 Ma) from the TimeTree database (http://www.timetree.org).

**Identification of tandem duplicated genes**. Tandem duplicated gene pairs were identified using duplicate_gene_classifier, a program of MCScanX. Functional classification of Gene Ontology (GO) categories was performed on the tandem duplicated genes using Blast2GO[95]. Fisher's exact test was used to calculate the statistical significance of enrichment ($p < 0.05$) with Benjamini–Hochberg's false discovery rate adjustment.

**Genome fractionation and subgenome dominance**. Genome fractionation in *O. kokonorica* was investigated using homoeologous gene pairs and singletons (whose duplicated copies were lost) in the two subgenomes that also had syntenic orthologs in *O. thomaeum*. Analysis of homoeolog expression bias was preformed following[21] for all five tissues (mature leaf, young leaf, root, rhizome, and rhizome tip) and samples under three abiotic treatments (see below). The five tissues were collected from the same place where plants were collected for genome sequencing.

**Genomic basis of divergence between *O. kokonorica* and *C. songorica***. In order to investigate the genomic basis of divergence between *O. kokonorica* and *C. songorica*, we compared the two species at the whole-genome level and the differentiation in gene expression under diverse abiotic stresses. To detect genomic variation between the two species, we first used NUCDIFF[96] to locate and categorize differences between the two species. The results were uploaded to RectChr (https://github.com/hewm2008/RectChr) for visualization. We also performed whole-genome comparisons using the nucmer (nucmer -maxmatch -l 100 -c 500) function from the MUMmer package. Structural variations (SVs) were called using Assemblytics[97] based on the output of nucmer. A custom Python script[98] was used to reformat the results from Assemblytics, and the effects of SVs (including high-, moderate-, low-, and modifier-impacted SVs) were annotated using SnpEff[99]. The genes categorized as highly-impacted-by-SV were subjected to GO enrichment and KEGG analyses.

To evaluate the differentiation in gene expression under drought, heat and cold treatments between the two species, seeds of *O. kokonorica* were collected from the same place where plants were collected for genome sequencing. Bleach-sterilized seeds were germinated and grown in a glasshouse under the same controlled conditions as for *C. songorica* used in ref. [17]. Four-

week-old seedlings were transplanted into individual pots with the same growth medium. Each pot was irrigated with 100 mL Hoagland nutrient solution every three days. We used the same conditions described in ref. [17] for the heat and cold treatment. Nine-week-old plants were treated under heat and cold (40 °C and 4 °C). For drought treatment, the soil water content was decreased to 10–20%, corresponding to light drought stress in ref. [17]. Shoots and roots of each plant were collected 24 h after treatment, immediately immersed in liquid nitrogen, then stored at −80 °C. Four biological replicates were used for each sample for transcriptomic analysis.

Total RNA was extracted using the Tiangen DP411 kit following the manufacturer's instructions. Messenger RNA (mRNA) was separated from the total RNA by Oligo (dT) and cleaved into short random fragments. Quality cDNA libraries were constructed by PCR enrichment and sequenced by BGI-Shenzhen Company (Wuhan, China) on the MGI2000 platform by $2 \times 150$ bp pair-end mode. Adapters, reads containing poly-N and lower-quality reads (<Q30) in the raw sequencing outputs were removed to obtain clean reads, which were then mapped to the *O. kokonorica* genome using HISAT2 v. 2.0.4[100]. We used StringTie v. 1.3.1[101] to calculate transcripts per million (TPM) of mRNA in each sample. TPM of genes was computed by summing the TPMs of transcripts in each gene group, and the DESeq2 R package[102] was used to perform differential expression analysis. Differentially expressed genes (DEGs) were defined as those having at least a twofold change in expression compared with those in controlled samples (false discovery rate, FDR < 0.05).

**Genomic basis of differentiation in floral development and rhizome growth between *Orinus* and *Cleistogenes***. To investigate the genomic basis of differentiation in floral development between *O. kokonorica* and *C. songorica*, we identified genes related to floral development (ABCDE model genes) in these two species based on the ABCDE models of *A. thaliana* and *O. sativa*. ABCDE genes of *A. thaliana* and *O. sativa* were downloaded from TAIR (https://www.arabidopsis.org/) and RICEDATA (https://ricedata.cn/), respectively. All genes of *O. kokonorica* and *C. songorica* were searched against the *A. thaliana* and *O. sativa* ABCDE model genes using BLASTP with an e-value cut-off of 1e −5. We then identified domains of candidate genes using HMMER v3.2.1[103] by searching the Pfam database[104]. Phylogenetic analysis was performed based on the ABCDE genes identified in *O. kokonorica*, *C. songorica*, and in *A. thaliana* and *O. sativa*. Only genes grouped with *A. thaliana* or *O. sativa* in the same clade were selected as orthologs.

We searched *BOP* genes in *O. kokonorica*, *C. songorica*, and *O. thomaeum* using the same strategy mentioned above with *A. thaliana* and *O. sativa BOP* genes. Domain confirmation and phylogenetic analyses were performed to identify orthologs.

**Statistics and reproducibility**. The functional enrichment analysis was performed using the ClusterProfile package, wherein the statistical significance of GO and KEGG terms was evaluated using Fisher's exact test combined with FDR correction for multiple testing ($p < 0.05$). For RNA-seq, four biological replicates were used for each tissue and each treatment.

## Data availability

The whole-genome sequencing data (including Illumina short reads, Nanopore long reads and Hi-C interaction reads), and RNA-seq data for the five tissues and three treatments have been deposited at NCBI under the Bioproject ID: PRJNA931908. The final assembled genome and genome annotations have been deposited in the National Genomics Data Center (https://bigd.big.ac.cn/?lang=en), under accession PRJCA018722. The source data behind the graphs in Figs. 2b and 3b are available as Supplementary

Data 1 and Supplementary Data 3–5, respectively. All other data are available from the corresponding author on reasonable request.

## Code availability

All scripts used in this study are available at https://github.com/qukunjing/Orinus-kokonorica[105].

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

## Acknowledgements

This work was supported by the Second Tibetan Plateau Scientific Expedition and Research (STEP) program (2019QZKK0502), the Fundamental Research Funds for the Central Universities (lzujbky-2021-ct14), the National Natural Science Foundation of China (31971391, 31960052) and the Program of Science and Technology International Cooperation Project of Qinghai Province (2022-HZ-802). We would like to thank the Big Data Computing Platform for Western Ecological Environment and Regional Development and Supercomputing Center at Lanzhou University for their support.

## Author contributions

G.P.R. conceived and designed the project. G.P.R. and J.Y.C. collected the materials. K.J.Q. and A.L. performed all the genomic analyses with the help of M.Y., W.J.M., S.W., H.Y.H., and X.S. Q.W.D. conducted the karyotyping analysis. K.J.Q. and G.P.R. wrote the manuscript. All the authors read and approved the manuscript.

## Competing interests

The authors declare no competing interests.
