## [Peer Review File · Communications Biology]

Reviewers' comments:

Reviewer #1 (Remarks to the Author):

Given the common adaptation of arid environments, the authors presented an interesting case study in relation to two closely related allotetraploid grasses of *Orinus kokonorica* and *Cleistogenes songorica* with divergent traits of cold/heat responses. *Orinus kokonorica* may serve as a valuable system for investigating impacts of WGD on genome evolution. In particular, *O. kokonorica* has experienced severe gene loss (large number of contracted gene families) but recent expansion of TEs compared with *C. songorica* during its adaptation to the high plateau. The language of the manuscript is good and it can be considered for publication after minor revisions as some points mentioned below.

1. Lines 40 and 51 why the older reference papers were placed later?
2. For the investigation of gene expression under different stress conditions, why not taking both *O. kokonorica* and *C. songorica* together and planting them under the same green house? Although you referenced the same conditions as those reported in Zhang et al. (2021), in fact it is difficult to keep the same environmental conditions which can further generate effects on gene expressions.
3. For LTRs, there is a biased occurrence between Gypsy and Copia elements. Should give some discussion on this phenomenon.
4. How about the expressions of those genes in relation to tandem repeats? In particular between subgenomes. Further details on this should be present.

Reviewer #2 (Remarks to the Author):

As an alpine perennial grass species endemic to the Qinghai-Tibet habitat in China, *Orinus kokonoricus* is ecologically very important. Resolving the genome of *O. kokonoricus* is the key to develop research revealing the genetic basis underlying its origin and subsequent ecological adaptations. In this study, Qu et al. reported the chromosome-scale assembly of *O. kokonoricus* genome, which is fundamental to understand the mechanisms underlying alpine adaptations. However, the work leaves open several questions.

First, necessary background is not mentioned. To what extent do flowers and longer rhizomes in *O. kokonoricus* contribute to its adaptation to the Qinghai-Tibet habitat?

Second, dimorphic flower is a very important adaptive trait, and development of dimorphic flower probably is due to the environment-responsive rewiring of genetic network. It is not clear that why authors chose to study ABCDE mode genes in the species *O. kokonorica* which only forms chasmogamous flowers (if correctly), since these genes are specifically involved in flower organ identity establishment and development.

Third, BOP genes are essential for rhizome tip stiffness, and loss of these genes leads to changes of rhizome morphology, such as severely twisted growth. However, influence of rhizomes on length is still clarified. Thus, BOP genes may be the only candidate to explore the potential mechanisms underlying longer rhizomes in *O. kokonorica*. Also, extra evidence should be provided to claim that more copies of BOP genes with high expressions are responsible for longer rhizomes in *O. kokonorica*.

line 385: The busco score seems a little bit low, considering a well annotated genome with score above 95%. What is the busco score of transcripts?

line 385: Gene losses play key roles during plant evolution. What categories are these genes enriched in?

line 399: Duplicated genes contribute to adaptation. It is not clear why tandem gene duplication was only mentioned here, without analysis of other types of duplicated genes.

line 426-427: Specify the tissues and stress treatments. For expression analysis in 5 tissues, how many replicates were used?

Reviewer #3 (Remarks to the Author):

This study reports the sequencing of a high quality genome for *Orinus kokonorica*, and this is reported as being an allotetraploid. Genome fractionation and gene homeolog expression bias are reported as being minimal. Changes with genes plausibly related to functional differences between this and a closely related species *Cleistogenes songorica*.

The introduction does a good job of giving the background of polyploidy and the species in question. The methods are generally good although some details are missing/unclear, see comments below. The main conclusions of the paper seem to be supported by the results, although clarification/further exploration is needed on the evidence for allopolyploidy and the lack of genome fractionation. The discussion of some points (e.g. the molecular dating/rate estimation) need to have the uncertainty in the estimates more thoroughly taken into account, and some smaller points go beyond the evidence presented in the paper and need to be scaled back.

Specific comments

Abstract

24-26: Only the fractionation bias and global homeolog expression in *O. kokonorica* are investigated in this paper.

27-30: This statement is quite vague, but I don't think anything has conclusively been identified as playing an important role in adaptive evolution/diversification, so this should be softened.

Introduction

68-70: This seems like a very strong claim (that WGD events should have contributed to these various things), and I'm not sure either of these references demonstrates this is the case. "May have contributed" would be fine.

101-106: References?

Methods

119: Is there a version number for this software?

228-229: Please give more details on how the rate is calculated. Is this a single rate calculated over the entire tree, or is it variable (and if so what model is being used). Is the fossil data used to calibrate the time the same as later in the paragraph? What is the uncertainty in the estimation of this rate - and how does this affect the dates associated with TE expansion?

255-257: Presumably this is where the young vs. mature leaf and root, rhizome tissue was harvested - this should be clarified.

Results

294: Typo between this line and Table S1 - is it 35.27 or 35.29?

300-302: Text and S8 says 97.6% complete, Table 1 says 90.60% complete? Which is it? I'm not sure why Table S2 is referenced here - perhaps the authors mean S8?

304-305: I would define "properly mapped" as properly paired, so according to Table S4 this should be 91.37% (or alternatively, just say the total mapped data).

313-315: Is this statistically significant? It looks like it could just be noise from Figure S3.

324-325: This differs from Table S8 and lines 300-302, (but matches Table 1).

333-335: Without demonstrating it statistically I think it's best avoided saying there's an inverse correlation with gene density - you could still say transposable elements were mainly distributed across the pericentric regions, while genes were mainly enriched in the more distal chromosomal regions, which seems plausible from Figure 1B (although a statistical demonstration of the relationship of LTR vs. gene density per Xbp would be better).

337-342: I'm not sure what the result estimated by Smudgeplot means here, as opposed to the k-mer analysis. I would have thought the high degree of colinearity among the 20 assembled pseudochromosomes is consistent with autotetraploidy as well as allotetraploidy? It may be worth splitting this section into the evidence for tetraploidy, then the evidence for allotetraploidy. It would also be worth adding a sentence explaining why a smaller peak at doubled multiplicity of the major peak implies allopolyploidy. Furthermore, considering the evidence of allopolyploidy in *C. songorica* in Zhang et al. 2021 seems to rely on a very similar set of analyses to the present paper, I am not sure this is necessarily an independent source of evidence. Perhaps it would be worth the authors expanding on how the additional analysis from Zhang et al. 2021 convincingly demonstrates allotetraploidy, this would make this point stronger. It would also be good to explain in more detail how the Ks results demonstrate allopolyploidy rather than autopolyploidy.

346: As far as I can tell from 198-200, the genomes were partitioned into subgenomes A and B and then Ks values calculated; so I'm not sure how the results of this can be used to justify partitioning the genomes.

369-374: What are the confidence intervals associated with these dates?

396-397: A bit too strong; this expansion could have happened before, or after, adaptation to the high plateau.

399-409: The fact that genes are tandemly duplicated is not evidence that they contributed to adaptation to the plateau environment. It is perhaps plausible, but not implied - and in any case the authors would need to explain why "leaf senescence", "aging" and "programmed cell death involved in cell development" and "defence response" are likely to be associated with adaptation to the plateau environment.

417-423: 14,598 out of 45,598 genes (~28% of the total) having loss of function seems high, but perhaps this is to be expected for tetraploids. It would be interesting to know how this compares to other species.

427-429: how many comparisons does this include? Are the different tissues compared only in the control treatment, or does this include every combination of tissues and treatments?

446-447: I am not sure how to interpret this figure, and whether it supports this statement. Where are the number of genes shown?

447-449: I am not sure what DEGs being 57 times greater means. There are 57 times as many (this doesn't look right from the previous line), they are expressed 57 times more highly? Or something else?

456-457: How are the genes having conserved functions defined? Not being differentially expressed in a given condition?

458-460: How is similarity defined - presumably not being differentially expressed, but this should be explicitly stated.

Discussion

488-492: I think this might be better in the introduction.

494: Where are the methods for the karyotyping?

502: "significantly" should be avoided when no statistical test is performed.

505-507: I think the uncertainty in the substitution rate and resulting uncertainty in the date of TE expansion is needed to link these two events convincingly.

516: Need to have confidence intervals of this date to assess this claim.

519: Table S3 should be referred to in relation to this claim in the results - ideally a statistical test would support this claim. From 418 it seems like a substantial proportion of genes are effectively pseudogenised by SVs; of these, do more occur on one of the subgenomes? This presumably would be a first step to genome fractionation.

532: "no substantial changes were found at the breakpoints" - I'm not sure what this means, or what results this is referring to.

543-544: I would say "in some flower development and rhizome growth related genes", as these are likely not all the genes involved.

553-556: The evidence in this paper is not sufficient to support this claim - you could say "may have contributed".

556-558: not clear what is meant by divergence here - if functional divergence, I don't think the evidence in the paper supports such a strong claim. I don't think there's any particular evidence given that these changes have played an important role in speciation.

569-570: This is not mentioned in the results - it should be prior to the discussion.

585-589: This makes sense; some of the above claims should be scaled back to reflect this. The paper would benefit from a conclusions section.

Figures: I'm sure this will be sorted out for the final publication but for the benefit of your reviewers, please don't make one figure much bigger than all the others so we have to zoom in/out all the time!

Table 1: The BUSCO score of this and the text disagree. Also the "repeats in genome" percentage from that given in the main text and Table S5.

Figure 1A: This is good.

Figure 3C: I'm not sure this figure conveys much meaningful information - I don't think the chromosomal distribution of DE genes is mentioned and the tracks are so small/faint it's hard to interpret much.

Figure 4C: Not clear what "Type" is referring to, presumably the SV types in B) but this would be worth clarifying.

Figure S10: typo - drought

Figure S11: Unclear what each of the individual labels (e.g. co_I1) corresponds to.

Figure S12: More explanation of this figure is needed. What does each row represent? What do the numbers correspond to?

Supplementary Figures;

Figure S2 - what are the units of the numbers plotted?

Figure S13 - this is not referred to in the main text.

Supplementary Tables;

Table S4:

I'm not clear how "regions" are defined. What "average depth (delete dup)" refers to should be clarified.

Table S15; What are the units for the numbers? What are the uncertainties associated with these?

How is "conserved function" or "changed functions" determined?

Reviewer #1 (Remarks to the Author):

Given the common adaptation of arid environments, the authors presented an interesting case study in relation to two closely related allotetraploid grasses of *Orinus kokonorica* and *Cleistogenes songorica* with divergent traits of cold/heat responses. *Orinus kokonorica* may serve as a valuable system for investigating impacts of WGD on genome evolution. In particular, *O. kokonorica* has experienced severe gene loss (large number of contracted gene families) but recent expansion of TEs compared with *C. songorica* during its adaptation to the high plateau. The language of the manuscript is good and it can be considered for publication after minor revisions as some points mentioned below.

Response: We thank the reviewer for this positive evaluation of our work and for putting forward many valuable comments and suggestions. Please be noted that we reorganized our revised manuscript by moving the section “Materials and Methods” to the end of the main text following the journal’s style.

1. Lines 40 and 51 why the older reference papers were placed later?

Response: In our initial version, the references were placed based on the first letter of last names. In the revised manuscript, we used the standard Nature referencing style.

2. For the investigation of gene expression under different stress conditions, why not taking both *O. kokonorica* and *C. songorica* together and planting them under the same green house? Although you referenced the same conditions as those reported in Zhang et al. (2021), in fact it is difficult to keep the same environmental conditions which can further generate effects on gene expressions.

Response: We agree that it is difficult to keep the same conditions which may further affect gene expression levels, but we think it will hardly change the expression patterns (e.g. from up-regulation to down-regulation). In this study, we aimed to examine the divergence of gene functions of the two species after WGD. We only focused on the orthologous genes that were both differentially expressed genes and only compared their differential expression patterns. In this way, we think we can avoid the effects generated from the two experiments. Therefore, we did not repeat the experiment for *C. songorica*. We clarified this clearer in the revised manuscript in the lines 296-301.

3. For LTRs, there is a biased occurrence between Gypsy and Copia elements. Should give some discussion on this phenomenon.

Response: To our knowledge, Gypsy occurs more abundance than Copia is a common phenomenon in plants. In *O. kokonorica*, we found that Gypsy is mainly enriched in centromeric regions, whereas Copia is distributed across the chromosomes in our species (result not shown). However, why Gypsy occurs more than Copia is not clear and not related to the aim of this study. Therefore, we decide to not add discussion on this phenomenon.

4. How about the expressions of those genes in relation to tandem repeats? In particular between subgenomes. Further details on this should be present.

Response: We do not quite understand this comment, but consider that the reviewer meant the tandemly duplicated genes. We investigated the expressions of these genes under cold, heat, and drought treatment. We found 734 (36.8%) out of 1,993 tandemly duplicated genes were differentially expressed in at least one tissue or treatment, and 236 (26.2%) out of the total 902 tandem gene clusters had at least two differentially expressed copies. We added this information in the lines 242-246 in the revised manuscript.

The tandemly duplicated genes are those arrayed very close across the same chromosome. Therefore, we cannot compare the expressions of them between subgenomes.

Reviewer #2 (Remarks to the Author):

As an alpine perennial grass species endemic to the Qinghai-Tibet habitat in China, *Orinus kokonoricus* is ecologically very important. Resolving the genome of *O. kokonoricus* is the key to develop research revealing the genetic basis underlying its origin and subsequent ecological adaptations. In this study, Qu et al. reported the chromosome-scale assembly of *O. kokonoricus* genome, which is fundamental to understand the mechanisms underlying alpine adaptations. However, the work leaves open several questions.

Response: We thank the reviewer for this evaluation of our work and for putting forward many valuable comments and suggestions. Please be noted that we reorganized our revised manuscript by moving the section “Materials and Methods” to the end of the main text following the journal’s style.

First, necessary background is not mentioned. To what extent do flowers and longer rhizomes in *O. kokonoricus* contribute to its adaptation to the Qinghai-Tibet habitat?

Response: Rhizomes in *O. kokonorica* can store and allocate nutrients for perennial growth and protect dormant buds underground for overwintering in the Qinghai-Tibet Plateau (QTP). With elongated rhizomes, *O. kokonorica* has developed a strong underground root system (Figure 1A), which may not only contribute to its high drought tolerance, but also has high ecological significance for sand fixation and water conservation in alpine arid regions.

We do not know if flowers in *O. kokonorica* have contributed to its adaptation to the QTP, but the dimorphic flowers in *C. songorica* are suggested to have contributed to its survival and reproductive success under drought conditions (Zhang et al., 2021).

We added these informations in the background in the lines 97-103.

Zhang, J. Y., Wu, F., Yan, Q., John, U. P., Cao, M. S., Xu, P., . . . & Wang, Y. R. (2021). The genome of *Cleistogenes songorica* provides a blueprint for functional dissection of dimorphic flower differentiation and drought adaptability. *Plant Biotechnology Journal*, 19(3), 532-547. doi:10.1111/pbi.13483

Second, dimorphic flower is a very important adaptive trait, and development of dimorphic flower probably is due to the environment-responsive rewiring of genetic network. It is not clear that why authors chose to study ABCDE mode genes in the species *O. kokonorica* which only forms chasmogamous flowers (if correctly), since these genes are specifically involved in flower organ identity establishment and development.

Response: We agree that dimorphic flower is a very important adaptive trait, and yes, *O. kokonorica* only forms chasmogamous flowers. Dimorphic flower is developed in *C. songorica*, and based on their transcriptomic data of *C. songorica* flowers, we found that many ABCDE genes showed higher expression in cleistogamous flowers (CLs) compared with that in chasmogamous flowers (CHs). These genes may play an important role during the development of CLs. Therefore, to investigate the genomic basis of differentiation in floral development between *O. kokonorica* and *C. songorica*, we chose to study ABCDE mode genes in the two species. We clarified this in the lines 313-318 in the revised manuscript.

We also agree that the development of dimorphic flower probably is due to the environment-responsive rewiring of genetic network. However, our present data which involves mainly genome and transcriptomic sequences of a species that only forms chasmogamous flowers are not appropriate to investigate this hypothesis. Further studies involving more data and experimental validation for species that have dimorphic flowers are needed to explore the genetic mechanism of the development of dimorphic flowers.

Third, BOP genes are essential for rhizome tip stiffness, and loss of these genes leads to changes of rhizome morphology, such as severely twisted growth. However, influence of rhizomes on length is still clarified. Thus, BOP genes may be the only candidate to explore the potential mechanisms underlying longer rhizomes in *O. kokonorica*. Also, extra evidence should be provided to claim that more copies of BOP genes with high expressions are responsible for longer rhizomes in *O. kokonorica*.

Response: Thanks for the comment. We agree that whether more copies of BOP genes with high expressions are responsible for longer rhizomes in *O. kokonorica* is needed for further verification. To do so, perhaps rhizome development transcriptomics, gene editing system and molecular experiments in *O. kokonorica* should be explored, which is however not possible at this stage. Future studies involving these new data and technologies would be extremely interesting to explore the potential mechanisms of BOP genes underlying longer rhizomes in *O. kokonorica*.

line 385 (should be line 325?): The busco score seems a little bit low, considering a well annotated genome with score above 95%. What is the busco score of transcripts?
Response: This 90.60% of BUSCO score is the evaluation of the annotated gene set. For the genome assembly, the BUSCO score is 97.6%, which is high. The relatively low BUSCO score for the annotated gene set may be due to the massive gene losses during the evolution of this species or incomplete gene prediction. The BUSCO score of transcripts is 77.8%, which is low and not mentioned in the text.

line 385: Gene losses play key roles during plant evolution. What categories are these genes enriched in?

Response: The contracted gene families are mainly enriched in normal GO terms, such as “transport, cell growth, secondary, metabolic process, response to endogenous stimulus, signal transduction, cell communication, response to biotic stimulus”, which are not mentioned in the main text.

line 399: Duplicated genes contribute to adaptation. It is not clear why tandem gene duplication was only mentioned here, without analysis of other types of duplicated genes.

Response: Tandem duplication is considered to be an important way for tetraploidy to continuously produce additional gene copies in addition to WGD (Session et al., 2016). Tandemly arrayed genes are thought to be volatile after gene duplication, and therefore the retained tandemly genes may indicate functional importance (Wang et al., 2021). Therefore, we focused on the tandemly duplicated genes and investigated their GO enrichment and expressions under three stress treatments (cold, heat, and drought). The results indicated that some GO terms related to adaptation to the plateau environment were enriched, and 734 (36.8%) out of 1,993 tandemly duplicated genes were differentially expressed in at least one tissue or treatment, suggesting that the tandemly duplicated genes may have contributed to adaptation to harsh plateau environment. We added this information in the lines 236-246 in the revised manuscript.

Session, A. M., Uno, Y., Kwon, T., Chapman, J. A., Toyoda, A., Takahashi, S., ... & Rokhsar, D. S. (2016). Genome evolution in the allotetraploid frog *Xenopus laevis*. *Nature*, 538(7625), 336-343.

Wang, X., Liu, S., Zuo, H., Zheng, W., Zhang, S., Huang, Y., ... & Xu, Q. (2021). Genomic basis of high-altitude adaptation in Tibetan *Prunus* fruit trees. *Current Biology*, 31(17), 3848-3860.

line 426-427: Specify the tissues and stress treatments. For expression analysis in 5 tissues, how many replicates were used?

Response: Thanks for the comment. We used 4 replicates in each of the tissues and stress treatments. We specified the tissues and stress treatments (lines 255-256) in the revised manuscript.

Reviewer #3 (Remarks to the Author):

This study reports the sequencing of a high quality genome for *Orinus kokonorica*, and this is reported as being an allotetraploid. Genome fractionation and gene homeolog expression bias are reported as being minimal. Changes with genes plausibly related to functional differences between this and a closely related species *Cleistogenes songorica*.

The introduction does a good job of giving the background of polyploidy and the species in question. The methods are generally good although some details are missing/unclear, see comments below. The main conclusions of the paper seem to be supported by the results, although clarification/further exploration is needed on the evidence for allopolyploidy and the lack of genome fractionation. The discussion of some points (e.g. the molecular dating/rate estimation) need to have the uncertainty in the estimates more thoroughly taken into account, and some smaller points go beyond the evidence presented in the paper and need to be scaled back.

Response: We thank the reviewer for this evaluation of our work and for putting forward many valuable comments and suggestions. Please be noted that we reorganized our revised manuscript by moving the section “Materials and Methods” to the end of the main text following the journal’s style.

Specific comments

Abstract

24-26: Only the fractionation bias and global homeolog expression in *O. kokonorica* are investigated in this paper.

Response: Thanks for the comment. We changed the term “their subgenomes” to “the two subgenomes of *O. kokonorica*”.

27-30: This statement is quite vague, but I don’t think anything has conclusively been identified as playing an important role in adaptive evolution/diversification, so this should be softened.

Response: We agree, and this has been now softened.

Introduction

68-70: This seems like a very strong claim (that WGD events should have contributed to these various things), and I’m not sure either of these references demonstrates this is the case. “May have contributed” would be fine.

Response: We fully agree, and this has been changed to “may have contributed”.

101-106: References?

Response: We added references here in the revised manuscript.

Methods

119: Is there a version number for this software?

Response: Yes, we added the version number in the revised manuscript.

228-229: Please give more details on how the rate is calculated. Is this a single rate calculated over the entire tree, or is it variable (and if so what model is being used). Is the fossil data used to calibrate the time the same as later in the paragraph? What is the uncertainty in the estimation of this rate - and how does this affect the dates associated with TE expansion?

Response: The program uses a maximum likelihood method with tree file with topology in Newick format and DNA sequence alignment in Phylip format as input files to calculate an average nucleotide substitution rate. The divergence times and TE insertion times were estimated based on this calculated average rate. We clarified this in the lines 563-569 in the revised manuscript.

Yes, the fossil data used to calibrate the time the same as later in the paragraph.

255-257: Presumably this is where the young vs. mature leaf and root, rhizome tissue was harvested - this should be clarified.

Response: No. The five tissues were collected from the same place where plants were collected for genome sequencing. We clarified this in the revised manuscript in the lines 581-582. The shoots and roots under treatments were collected in greenhouse.

Results

294: Typo between this line and Table S1 - is it 35.27 or 35.29?

Response: 35.39 is the right value. We corrected it in the revised manuscript.

300-302: Text and S8 says 97.6% complete, Table 1 says 90.60% complete? Which is it? I'm not sure why Table S2 is referenced here - perhaps the authors mean S8?

Response: We apologize for the confusion. The two busco values represent different meanings, 97.6% is for the evaluation of the genome assembly and 90.60% is for the evaluation of the annotated genes. Table S2 here is a further explanation of contig N50, but to avoid confusion, we refer to Table S2 in advance.

304-305: I would define "properly mapped" as properly paired, so according to Table S4 this should be 91.37% (or alternatively, just say the total mapped data).

Response: Yes, we changed the text accordingly.

313-315: Is this statistically significant? It looks like it could just be noise from Figure S3.

Response: The LTR insert time is estimated based on the LTRs in each species. As the LTRs in the two species are different. We cannot compare whether the estimates differ significantly, but simply compare the results.

324-325: This differs from Table S8 and lines 300-302, (but matches Table 1).

Response: We apologize for the confusion. We got confused between the old version and the new version of the busco reference set. The results in the text are correct and we modified it in Table S8.

333-335: Without demonstrating it statistically I think it's best avoided saying there's an inverse correlation with gene density - you could still say transposable elements were mainly distributed across the pericentric regions, while genes were mainly enriched in the more distal chromosomal regions, which seems plausible from Figure 1B (although a statistical demonstration of the relationship of LTR vs. gene density per Xbp would be better).

Response: We fully agree with your comment, we modified the sentence (lines 151-152) in the revised manuscript.

337-342: I'm not sure what the result estimated by Smudgeplot means here, as opposed to the k-mer analysis. I would have thought the high degree of colinearity among the 20 assembled pseudochromosomes is consistent with autotetraploidy as well as allotetraploidy? It may be worth splitting this section into the evidence for tetraploidy, then the evidence for allotetraploidy. It would also be worth adding a sentence explaining why a smaller peak at doubled multiplicity of the major peak implies allopolyploidy. Furthermore, considering the evidence of allopolyploidy in *C. songorica* in Zhang et al. 2021 seems to rely on a very similar set of analyses to the present paper, I am not sure this is necessarily an independent source of evidence. Perhaps it would be worth the authors expanding on how the additional analysis from Zhang et al. 2021 convincingly demonstrates allotetraploidy, this would make this point stronger. It would also be good to explain in more detail how the Ks results demonstrate allopolyploidy rather than autopolyploidy.

Response: Thanks for the comment. Smudgeplot shows two k-mer peaks. The obvious peak of AABB type of k-mers may imply the allopolyploid origin of *O. kokonorica*.

We split this section into the evidence for tetraploidy then the evidence for allotetraploidy. We also clarified how the Ks values demonstrate allopolyploidy rather than autopolyploidy. We extensively modified the whole paragraph to clarify all these concerns (lines 154-168).

346: As far as I can tell from 198-200, the genomes were partitioned into subgenomes A and B and then Ks values calculated; so I'm not sure how the results of this can be used to justify partitioning the genomes.

Response: Sorry for the confusion. Ks is assumed to be neutral, if *O. kokonorica* is an autotetraploidy, we would expect that the Ks values between the two duplicated chromosomes of *O. kokonorica* and one of their two syntenic *C. songorica* chromosomes are similar. However, our result showed that each *C. songorica* chromosome corresponded with a pair of *O. kokonorica* pseudochromosomes with apparently different Ks values, supporting the allotetraploid origin of *O. kokonorica*. Then based on the Ks values between the *O. kokonorica* genome and the two *C.*

songorica subgenomes, we determined the two subgenomes of *O. kokonorica*. We also added two sentences to point out that the two subgenomes partitioned in *C. songorica* should be treated with caution, but based on their subgenomes to determine the subgenomes of *O. kokonorica* will not affect our comparative genomic analyses (lines 171-174).

369-374: What are the confidence intervals associated with these dates?

Response: We added the confidence intervals for these dates in the revised manuscript.

396-397: A bit too strong; this expansion could have happened before, or after, adaptation to the high plateau.

Response: We agree and removed this sentence in the revised manuscript.

399-409: The fact that genes are tandemly duplicated is not evidence that they contributed to adaptation to the plateau environment. It is perhaps plausible, but not implied - and in any case the authors would need to explain why “leaf senescence”, “aging” and “programmed cell death involved in cell development” and “defence response” are likely to be associated with adaptation to the plateau environment.

Response: Species at high altitudes are subject to genetically adaptive evolution that contributes to their survival in the harsh environment of prolonged exposure to high UV-B irradiation, low temperature and hypoxia. It is generally accepted that species at higher altitudes have longer leaf life than those at lower altitudes. For example, leaf lifespans of both subtropical evergreen broadleaf forests and conifers in the eastern part of the Gongga Mountains at 1,900-3,700 m altitude increase along the altitude gradient (Luo et al., 2006). We added this argument in the revised manuscript (lines 241-242). To avoid confusion, the “aging”, “programmed cell death involved in cell development” and “defence response” were removed in the revised version.

We also investigated their expressions under three stress treatments (cold, heat, and drought). The results indicated that 734 (36.8%) out of 1,993 tandemly duplicated genes were differentially expressed in at least one tissue or treatment, suggesting that the tandemly duplicated genes may have contributed to adaptation to the harsh plateau environment. We added this information in the lines 242-246 in the revised manuscript.

Luo, J., Zang, R., & Li, C. (2006). Physiological and morphological variations of *Picea asperata* populations originating from different altitudes in the mountains of southwestern China. *Forest Ecology and Management*, 221(1-3), 285-290.

417-423: 14,604 out of 45,598 genes (~32% of the total) having loss of function seems high, but perhaps this is to be expected for tetraploids. It would be interesting to know how this compares to other species.

Response: These genes are not loss of function. They are categorized as SV-high-impact genes. SVs around these genes may have a high or disruptive impact on the proteins. SVs between closely related species seem to have impact on a relatively high

proportion of genes. For example, in our previous paper, SVs have impact on 18,604 (~44%) out of 42,066 genes between two *Medicago truncatula* ecotypes (Li et al., 2022).

Li, A., Liu, A., Wu, S., Qu, K., Hu, H., Yang, J., ... & Ren, G. (2022). Comparison of structural variants in the whole genome sequences of two *Medicago truncatula* ecotypes: Jemalong A17 and R108. *BMC Plant Biology*, 22(1), 1-15.

427-429: how many comparisons does this include? Are the different tissues compared only in the control treatment, or does this include every combination of tissues and treatments?

Response: All the 5 tissues and 3 treatments were used for homoeolog expression dominance analysis. For each one, we got a result. Please see Figure S10 for details.

446-447: I am not sure how to interpret this figure, and whether it supports this statement. Where are the number of genes shown?

Response: We added the number of DEGs in shoot and root under each of the three treatments.

447-449: I am not sure what DEGs being 57 times greater means. There are 57 times as many (this doesn't look right from the previous line), they are expressed 57 times more highly? Or something else?

Response: We apologize for the confusion. Here we meant the number of DEGs in roots were about 57 times greater than in shoots under drought treatment. We clarified this in the revised manuscript.

456-457: How are the genes having conserved functions defined? Not being differentially expressed in a given condition?

Response: No, the conserved function was defined as those orthologous DEG pairs have the same differential expression pattern in the same tissue under the same treatment. We clarified this in the lines 304-305 in the revised manuscript.

458-460: How is similarity defined - presumably not being differentially expressed, but this should be explicitly stated.

Response: Thanks. It should be "the same expression patterns". We changed it in the revised manuscript.

Discussion

488-492: I think this might be better in the introduction.

Response: We moved it to the introduction.

494: Where are the methods for the karyotyping?

Response: Another group helped us for karyotyping. There is no method for this analysis. To avoid confusion, we changed the sentence to “The karyotyping result together with our comparative genomic analyses...”.

502: “significantly” should be avoided when no statistical test is performed.

Response: Done

505-507: I think the uncertainty in the substitution rate and resulting uncertainty in the date of TE expansion is needed to link these two events convincingly.

Response: The insertion of LTRs is a dynamic process. We calculated the insertion time for each LTR based on a single substitution rate. Then plotted all the estimated times to obtain the result of Figure S3. The peak shows the time when the LTR inserted most actively. Therefore, we think the link of the LTR expansion and glaciation in the QTP is convincing.

516: Need to have confidence intervals of this date to assess this claim.

Response: The confidence intervals for the polyploidization times of *E. tef* and *L. chinensis* were not mentioned in their original articles.

519: Table S3 should be referred to in relation to this claim in the results - ideally a statistical test would support this claim. From 418 it seems like a substantial proportion of genes are effectively pseudogenised by SVs; of these, do more occur on one of the subgenomes? This presumably would be a first step to genome fractionation.

Response: Thanks for the comment. We did do genome fractionation analysis in our initial version. The result was shown in Figure S13, which was unfortunately not cited in the initial version, but mentioned in the discussion (now in lines 372-373). We added one section “Genomic fractionation and subgenome dominance bias” in both Results (lines 274-252) and Materials and Methods (lines 576-582) in the revised manuscript to clarify this.

As replied before, the SV-high-impact genes are not loss of function. So, we did not perform more analysis on these genes.

532: “no substantial changes were found at the breakpoints” - I’m not sure what this means, or what results this is referring to.

Response: It means there are no changes for genes near the breakpoint between the two species. We modified the term “substantial” to “gene” in the revised manuscript.

543-544: I would say “in some flower development and rhizome growth related genes”, as these are likely not all the genes involved.

Response: Done.

553-556: The evidence in this paper is not sufficient to support this claim - you could say “may have contributed”.

Response: Done.

556-558: not clear what is meant by divergence here - if functional divergence, I don't think the evidence in the paper supports such a strong claim. I don't think there's any particular evidence given that these changes have played an important role in speciation.

Response: We meant genomic divergence here. We clarified this in the revised manuscript.

569-570: This is not mentioned in the results - it should be prior to the discussion.

Response: We now mentioned it in the results.

585-589: This makes sense; some of the above claims should be scaled back to reflect this. The paper would benefit from a conclusions section.

Response: We added a Conclusions section in the revised manuscript.

Figures: I'm sure this will be sorted out for the final publication but for the benefit of your reviewers, please don't make one figure much bigger than all the others so we have to zoom in/out all the time!

Response: Sorry for the inconvenience. We modified the sizes of figures.

Table 1: The BUSCO score of this and the text disagree. Also the "repeats in genome" percentage from that given in the main text and Table S5.

Response: We are sorry for the confusion. The values in the text are correct. We corrected the wrong values in Table 1, Table S5 and Table S8, and checked all other numbers and values throughout the manuscript.

Figure 1A: This is good.

Figure 3C: I'm not sure this figure conveys much meaningful information - I don't think the chromosomal distribution of DE genes is mentioned and the tracks are so small/faint it's hard to interpret much.

Response: Thanks for the comment. However, we think Figure 3C is necessary to support our claim that most orthologous genes have different differentially expressed patterns between the two species. For example, the orthologous genes in the two B1 chromosomes displayed clearly different colors (blue color indicates downregulation, red color indicates upregulation) under each treatment, indicating different differentially expressed patterns.

Figure 4C: Not clear what "Type" is referring to, presumably the SV types in B) but this would be worth clarifying.

Response: Yes, "Type" is the SV types in B. We clarified it in the legend.

Figure S10: typo – drought

Response: Done.

Figure S11: Unclear what each of the individual labels (e.g. co_I1) corresponds to.

Response: ck: Control Check; he: heat treatment; dr: drought treatment; co: cold treatment. r: root; s: shoot; 1, 2, 3, 4 mean the replicates of treatments. We clarified this in the legend of this figure.

Figure S12: More explanation of this figure is needed. What does each row represent? What do the numbers correspond to?

Response: Thanks for the comment. We added more explanation for this figure.

Figure S2 - what are the units of the numbers plotted?

Response: The legend on the right shows all interactions between each pixel point and another pixel point in the left panel. The numbers correspond to the number of reads.

Figure S13 - this is not referred to in the main text.

Response: This figure is now referred in the revised manuscript.

Supplementary Tables;

Table S4:

I'm not clear how "regions" are defined. What "average depth (delete dup)" refers to should be clarified.

Response: It is "genomic regions".

"Average depth (delete dup)" means the average depth after deleting haplotigs and overlaps in an assembly based on read depth. We added this below the table.

Table S15; What are the units for the numbers? What are the uncertainties associated with these? How is "conserved function" or "changed functions" determined?

Response:

- (i) The numbers represent the $\text{Log}_2(\text{Fold change})$ values between the control and treatment. They are ratios of $\text{TPM}_{\text{treatment}}/\text{TPM}_{\text{control}}$, so there are no units.
- (ii) They are ratios, so no uncertainties.
- (iii) The "conserved function" was defined as those orthologous DEG pairs have the same differential expression pattern in the same tissue under the same treatment between the two species. If the pattern changes, we defined as "changed functions"

Reviewers' comments:

Reviewer #1 (Remarks to the Author):

no further comments.

Reviewer #2 (Remarks to the Author):

I have read the revised manuscript "Comparative genomics provides insights into the origin, adaptive evolution and further diversification of two closely related grass genera" by Qu et al. This version of the manuscript is much improved, with only one issue yet to be addressed.

It is widely acknowledged ABCDE genes are involved in organ identity establishment and development of flowers. Qu et al. attempted to give a plausible explanation with inferred literature for analysis of ABCDE genes, aiming to provide a clue to understand the potential mechanism underlying formation of chasmogamous and cleistogamous flowers in studied species, however, it is still not well justified. The reason should be further explained with evidence and/or hypothesis that differential expression and/or loss of ABCDE gene orthologues lead to formation of chasmogamous and cleistogamous flowers.

Reviewer #4 (Remarks to the Author):

The revised manuscript satisfactorily addressed the issues raised by previous reviewers. I do not have further comments.

Responses to reviewers' comments

Reviewer #2 (Remarks to the Author):

I have read the revised manuscript “Comparative genomics provides insights into the origin, adaptive evolution and further diversification of two closely related grass genera” by Qu et al. This version of the manuscript is much improved, with only one issue yet to be addressed.

It is widely acknowledged ABCDE genes are involved in organ identity establishment and development of flowers. Qu et al. attempted to give a plausible explanation with inferred literature for analysis of ABCDE genes, aiming to provide a clue to understand the potential mechanism underlying formation of chasmogamous and cleistogamous flowers in studied species, however, it is still not well justified. The reason should be further explained with evidence and/or hypothesis that differential expression and/or loss of ABCDE gene orthologues lead to formation of chasmogamous and cleistogamous flowers.

Response: Thank you for the comment. A recent study has proposed a putative model for chasmogamous and cleistogamous flower development in *Amphicarpaea edgeworthii*: The specifically morphological characters of petals, stamens, and ovules of the CL flowers may be attributed to the depressed expression of A/B/E-class genes and the enhanced expression of isoforms in C-class during the development process (see attached figure). Zhang et al. (2021) also found that overexpression of *CsAP2_9* gene confers an abnormal palea in a spikelet and shows degenerated lodicules in flowers in *C. songorica*. Based on these two literatures, we speculate that the differential expression of the ABCDE genes may contribute to the development of the CH and CL flowers. In our previous version, we only focused on the ABCDE genes that had higher expression in CLs compared with that in CHs in *C. songorica*. In this revised manuscript, we also investigated the genes that had lower expression (two-fold change) in CLs compared with that in CHs in *C. songorica*, which resulted in 3 more genes. One of the 3 genes was lost in *O. kokonorica* (Figure 4a blue stars) and many SVs were detected between the other 2 orthologues of the two species (Figure 4c). All these information were added in the revised manuscript (lines 318-326; lines 331-335; lines 426-432).

Whether the losses of ABCDE gene orthologues lead to formation of CH and CL flowers is still not clear. Nevertheless, because the two copies in *C. songorica* are both differentially expressed genes, we speculate the loss of one copy in *O. kokonorica* may lead to different gene expression, which may affect the development of flower.

We have also pointed out in our previous version that because our present data are not appropriate to gain detailed mechanistic explanations of the flower development in the two species. Further studies involving more comparative genomic analyses associated with experimental validation would be needed to enhance our understanding of the molecular mechanisms of flower development in the two genera (lines 442-445).

Figure The CH and CL flowering ABCDE model that specifies floral organs is proposed based on the gene expression values (bar heights) in *Amphicarpaea edgeworthii* (Liu et al., 2021).

Liu, Y., Zhang, X., Han, K., Li, R., Xu, G., Han, Y., ... & Wan, S. (2021). Insights into amphicarpity from the compact genome of the legume *Amphicarpaea edgeworthii*. *Plant Biotechnology Journal*, 19(5), 952-965.

Zhang, J., Wu, F., Yan, Q., John, U. P., Cao, M., Xu, P., ... & Wang, Y. (2021). The genome of *Cleistogenes songorica* provides a blueprint for functional dissection of dimorphic flower differentiation and drought adaptability. *Plant biotechnology journal*, 19(3), 532-547.